# Mucosal tenofovir 1% gel stimulates cell proliferation and type I/III interferon pathways

Sean M. Hughes,[1] Fernanda L. Calienes,[1] Claire N. Levy,[1] Urvashi Pandey,[1] Germán G. Gornalusse,[1] Ross D. Cranston,[2] Javier R. Lama,[3] Jim Pickett,[4] Rhonda M. Brand,[5,6] Craig W. Hendrix,[7] Mark A. Marzinke,[7] James Y. Dai,[8] Bhavna Balar,[9] Jessica E. Justman,[10,11] Gonasagrie Nair,[12,13] Alicia R. Berard,[14,15] Kenzie Birse,[16] Laura Noel-Romas,[15,17] Kenneth H. Mayer,[18,19,20] Joanne D. Stekler,[21,22,23] Romel Mackelprang,[1] Adam D. Burgener,[15,17,24] Ian McGowan,[6] Cheryl M. Cameron,[25] Mark J. Cameron,[26] Kim A. Woodrow,[27] Florian Hladik[1,28,29]

**ABSTRACT**  Oral tenofovir is a key antiretroviral used for treatment and pre-exposure prophylaxis (PrEP) of human immunodeficiency virus (HIV). A gel form has been tested for vaginal and rectal PrEP. We have shown that 7 days of tenofovir 1% gel had broad-ranging effects on gene expression in the rectum, especially suppression of anti-inflammatory mediators and induction of cell proliferation. Similarly, oral PrEP induced type I/III interferon-stimulated genes in the gut. It is unknown how long these effects last and whether they occur in other relevant body compartments. We measured the transcriptomes and proteomes of tissue samples obtained before and after daily topical tenofovir 1% gel application for 14 days (Microbicide Trials Network [MTN]-014 trial, rectal and vaginal) or 56 days (MTN-017 trial, rectal). While many changes seen after 7 days diminish after 14 and 56 days, some remain, notably increases in cell proliferation- and type I/III interferon-related genes. Vaginal gel uniquely induces changes related to epithelial-mesenchymal transition and angiogenesis. Induction of type I/III interferon-related genes is the most consistent and persistent mucosal response to tenofovir, occurring after both oral and topical use and at all tested time points. Hypothetically, interferon induction could improve antiviral efficacy, but also contribute to an increased chronic disease burden in people with HIV.

**IMPORTANCE** Analyzing gene expression data from three separate clinical trials, we find that the antiretroviral drug tenofovir, which belongs to the class of nucleotide analogue reverse transcriptase inhibitors, induces the type I/III interferon system of innate immunity in the mucosa. This effect occurs in the absence of HIV infection and manifests itself over various treatment durations and after both oral and topical drug delivery. Tenofovir and other related medications are important components of long-term antiretroviral treatment taken by people living with HIV. Therefore, this unexpected immunological effect might need to be considered as a potential contributor to comorbidities in people living with HIV, as well as an immunopharmacological co-factor when testing novel HIV cure interventions.

**CLINICAL TRIALS** This study is registered with ClinicalTrials.gov as NCT01768962, NCT01687218, and NCT01232803.

**KEYWORDS**  microbicide, antiretroviral treatment, pre-exposure prophylaxis, tenofovir, HIV

The nucleotide reverse transcriptase inhibitor (NRTI) tenofovir is a mainstay drug of antiretroviral therapy (ART) and of oral pre-exposure prophylaxis (PrEP) for human

**Peer Reviewer** Arjit Vijay Jeyachandran, University of California, Los Angeles, Los Angeles, California, USA

Address correspondence to Florian Hladik, florian@uw.edu.

The authors declare no conflict of interest.

See the funding table on p. 23.

immunodeficiency virus (HIV) (1–4). It is also being tested as topical PrEP administered to the vagina or the rectum to prevent sexual HIV transmission. In our previous paper in this journal, we described how daily rectal application of tenofovir 1% gel for 7 days caused a broad spectrum of gene expression changes in the rectum, particularly changes suggesting increased inflammation, enhanced cellular proliferation, and mitochondrial toxicity (5). These aforementioned observations were derived from the Microbicide Trials Network (MTN)-007 trial (6).

Elsewhere, we have described the effects of tenofovir disoproxil fumarate/emtricitabine (TDF/FTC) pills, taken orally for 2 months as daily HIV PrEP, on gene expression in the rectum and duodenum in the absence of HIV (7). TDF is a prodrug of tenofovir and is used for oral administration, while tenofovir itself is used in topical gel formulations. FTC belongs to the same NRTI class of antiretroviral drugs as tenofovir. After 2 months of daily oral TDF/FTC use, we observed an increase of type I/III interferon (IFN)-stimulated genes (ISGs) in intestinal tissue samples. These findings were consistent across three independent clinical trials, measured by several methodologies at the gene and protein level, and across two anatomical sites in the gut, duodenum, and rectum. Thus, we also became interested to know whether we could identify this ISG-stimulating effect after topical tenofovir use for more than 7 days.

In the current analyses, we evaluated samples from two additional studies: MTN-014, a 14-day crossover trial evaluating the pharmacokinetics of tenofovir 1% gel in the rectum and vagina (14-day treatment in each compartment) (8), and MTN-017, a randomized sequence safety and acceptability trial of daily oral TDF/FTC and daily and pericoital rectal tenofovir 1% gel (56 days of treatment in each of the three study periods) (9). Our new findings address key remaining areas of interest: gene expression changes in the rectum with longer-term or pericoital use of the gel, gene expression changes in the vagina with vaginal use, and cross-compartmental gene expression changes (i.e., in the vagina after rectal gel application and *vice versa*). Lastly, we look at the induction of the type I/III interferon system observed after oral TDF/FTC use and assess whether it also occurs during longer-term topical tenofovir use.

Our data is relevant for real-life use of topical tenofovir 1% gel as an HIV prophylactic in the rectum or vagina and may also bear relevance on how to best use other topical microbicides, such as a tenofovir douche (10) or a gel or ring containing dapivirine (11–14). Furthermore, finding commonalities in mucosal changes between oral and topical tenofovir use would underscore the importance of considering these effects in both HIV prevention and treatment settings.

## RESULTS

### Samples/studies/overview

Here, we present new gene expression data from topical tenofovir 1% gel use in the MTN-014[8] and MTN-017[9] trials. These trials were performed among participants without HIV who used the study products as PrEP. MTN-014 was a prospective crossover trial in which participants completed two study periods, each 14 days in length. One study period consisted of daily application of reduced-glycerin tenofovir 1% gel application to the vagina, and the other consisted of daily application of the gel to the rectum. Participants were randomized to start with either rectal or vaginal application, with a 6-week washout period between study periods. Rectal and vaginal tissue samples were collected at three time points: enrollment, 24 h after the final gel application of study period 1, and 24 h after the final gel application of study period 2. Specimen collection was timed to coincide with steady-state tenofovir concentrations in the tissues. Samples were available from 12 participants at both anatomic sites at each of the three time points (72 total samples; Table 1).

MTN-017 was a prospective double crossover trial with three study periods: 56 days of daily reduced-glycerin tenofovir 1% gel application to the rectum, 56 days of pericoital gel application to the rectum, and 56 days of oral use of TDF/FTC. Participants were randomly assigned to a sequence of product use, and all participants completed all

**TABLE 1** Characteristics of studies from which samples were obtained[a]

| Study | Biological sex | Treatment length | Treatment interval | Treatment location | Sample location | Sample number |
|---|---|---|---|---|---|---|
| MTN-014 NCT01768962 (8) | Female | 14 days | Daily | Rectum | Rectum | 12 |
| | | | | | Vagina | 12 |
| | | | | Vagina | Rectum | 12 |
| | | | | | Vagina | 12 |
| MTN-017 NCT01687218 (9) | Male | 56 days | Daily | Rectum | Rectum | 36 |
| | | | Pericoital | | | 36 |
| | | | Daily | Oral TDF/FTC | Rectum | 36 |
| MTN-007 NCT01232803 (6) | Male | 7 days | Daily | Rectum | Rectum | 8 |

[a]Unshaded rows indicate the source of samples for data newly reported in this study, while shaded rows indicate the source of samples previously reported, but included here for context. All three studies examined the effects of the reduced-glycerin formulation of tenofovir 1% gel. Study sizes listed in the table represent the number of participants providing specimens for the gene expression studies reported, not the number of participants in each clinical study itself. Biological sex was presumed based on anatomical organs. Baseline samples were collected from each study prior to treatment ($n = 12$ rectum and $n = 12$ vagina from MTN-014, $n = 36$ rectum from MTN-017, and $n = 8$ rectum from MTN-007).

study periods. In the pericoital gel application study period, participants were instructed to apply the gel before and after receptive anal intercourse, with no more than two applications in 24 h. As previously reported, participants used an average of 2.9 doses per week (standard deviation 1.6) during the pericoital study period (15). There were 1-week washout periods between study periods. The data about oral use have been reported in detail previously (7). Participants provided rectal biopsy samples at four time points: enrollment and at the end of each of the three study periods. Samples were available from 36 participants at each time point (144 total samples; Table 1).

To briefly recapitulate from our prior reports (5, 6), MTN-007 was a randomized, double-blind, placebo-controlled study of reduced-glycerin tenofovir 1% gel. For the present article, we assessed gene expression changes in 15 cm rectal biopsy samples obtained from eight participants, comparing their baseline samples to their matched samples obtained after seven daily applications of the gel.

## Most gene expression changes wane with longer-term rectal use of tenofovir 1% gel

In our previous report, we observed substantial gene expression changes in rectal tissue samples associated after seven days of daily rectal tenofovir 1% gel use (MTN-007). To determine whether those gene expression changes persisted after longer use, we performed gene expression microarrays on rectal tissue samples collected after 14 days (MTN-014) and 56 days of daily rectal tenofovir 1% gel use (MTN-017). In each study, we determined gene expression changes by comparing each participant's tissue sample to their own pre-treatment tissue sample. Because no samples were available from a single study tracking tenofovir 1% gel use longitudinally over time, we used a correlation-based approach across these three studies to assess the long-term effects of tenofovir use.

We first compared the patterns of gene expression changes between MTN-007 and the new studies. Specifically, we evaluated correlations between the log2-fold changes for each gene calculated in MTN-007 after 7 days of gel use and the log2-fold changes of the same gene after 14 and 56 days of gel use (Fig. 1). The rationale behind these correlation tests was that a strong correlation between gene expression changes from MTN-007 (7 days) and changes in MTN-014 (14 days) and MTN-017 (56 days) would indicate that tenofovir 1% gel use causes lasting and consistent changes in gene expression over time. In contrast, if there were no correlations, that would suggest that tenofovir 1% gel use did not cause consistent changes in gene expression over time. When considering all 15,591 genes detectable in all three studies, we observed significant, but weak, correlations between the fold changes after 7 days and the fold changes after 14 and 56 days (Fig. 1A).

We then repeated this analysis on only those 1,226 genes that were differentially expressed after 7 days of use in MTN-007 and were detectable in MTN-014 and MTN-017.

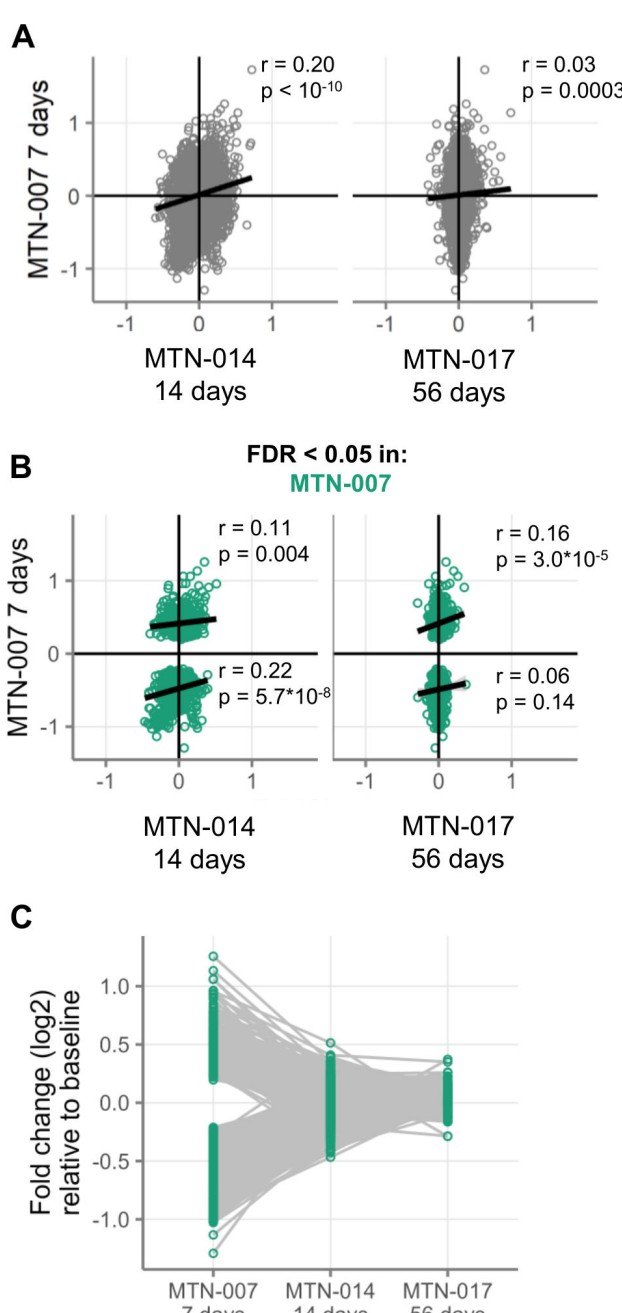

**FIG 1** Waning over time of gene expression changes in rectal tissue samples after daily topical 1% tenofovir application to the rectum. Rectal biopsies in all three studies were taken ~15 cm from the anus at baseline and after 7 days (MTN-007), 14 days (MTN-014), or 56 days (MTN-017) of daily gel application. (A) Correlation of gene expression changes between those observed in the previous study (MTN-007) after 7 days of use and each of the new studies. Each point represents the log2-fold change of a single gene from the studies indicated on the axes. All detectable genes are shown. Positive fold changes mean higher expression during product use, and negative fold changes mean higher expression at baseline. (B) The same plot as in panel A, except limited to only those genes that were differentially expressed (FDR < 0.05) after 7 days of gel use in MTN-007. (C) The magnitude of fold changes of those genes that were differentially expressed after 7 days of gel use in MTN-007.

Of these genes, none were differentially expressed in MTN-014, and 25 were differentially expressed in MTN-017. Given the varying sample sizes—and therefore statistical power —between the studies, we assessed similarities in fold changes between studies without

filtering for differential expression in each study. To account for the bimodal distribution of differentially expressed genes, we calculated the correlations separately for positive and negative fold changes in MTN-007. We again observed significant correlations between the log2-fold changes after 7 days and the changes after 14 and 56 days (Fig. 1B).

In addition to the direction of regulation, we also assessed the magnitudes of fold changes at 7, 14, and 56 days compared to the respective pre-treatment expression levels for those genes that were differentially expressed after 7 days of daily tenofovir 1% gel use in MTN-007. The magnitudes of these expression changes decreased substantially over time (Fig. 1C). For each day of gel use, the average absolute fold change decreased by 0.006 (linear mixed-effects model, $P < 10^{-10}$), implying a complete waning of the effect 74 days following the first time point.

Thus, we observed correlated gene expression changes with the use of topical 1% tenofovir gel across three independent trials of varying treatment durations. However, these correlations and the magnitudes of gene expression changes diminish over time. This finding suggests that topical tenofovir's effects wane with continued longer-term use.

## Some gene expression changes persist with longer-term rectal use of tenofovir 1% gel

Despite this consistent waning of global gene expression changes over time, we still did observe significant correlations in gene expression between the changes after 7 days and the changes after 14 and 56 days (Fig. 1A and B). To characterize these more durable changes, we first assessed whether there were any differentially expressed genes (FDR < 0.05) at 14 and 56 days. As shown in Table 2, we found no differentially expressed genes (defined by false discovery rate < 0.05) in any arm of MTN-014 (two weeks of gel use). In MTN-017, we found 108 upregulated and 65 downregulated genes after two months of daily use. Of these 173 genes, 25 were differentially expressed in both MTN-007 and MTN-017, with 23 having fold changes in the same direction in both studies. File S1 lists the fold changes and FDR-adjusted $P$-values for all genes.

To assess the durability of the changes detected in MTN-017 at day 56, we correlated the 173 genes differentially expressed after 56 days of daily tenofovir 1% gel rectal use with the changes of these same genes at day 7 (MTN-007) (Fig. 2A, right panel). We also assessed these 173 genes (65 downregulated and 108 upregulated) for their correlation between day 7 (MTN-007) and day 14 (MTN-014) (Fig. 2A, left panel). In contrast to the genes from MTN-007 discussed above, for these 173 genes from the MTN-017 data set, the correlation between their day 14 (MTN-014) and their day 7 (MTN-007) changes was weaker than the correlation between their day 56 (MTN-017) and their day 7 (MTN-007) changes. There was also only a very slow waning in the magnitude of gene expression changes over time for these genes (Fig. 2B). For every day of gel use, the average absolute fold change decreased by 0.0008 (linear mixed-effects model, $P = 0.0014$), approximately 7.6 times slower than the waning of the MTN-007 genes (Fig. 1C).

We next assessed the gene expression data from each study, MTN-007, MTN-014, and MTN-017, separately to examine changes in larger biological functions using all of the

**TABLE 2** Differentially expressed genes[a]

| Study | Treatment length | Treatment interval | Treatment location | Sample | Genes up | Genes down | Total |
|---|---|---|---|---|---|---|---|
| MTN-007 | 7 days | Daily | Rectum | Rectum | 655 | 599 | 20,869 |
| MTN-014 | 14 days | Daily | Rectum | Rectum | 0 | 0 | 17,526 |
| MTN-014 | 14 days | Daily | Rectum | Vagina | 0 | 0 | 17,542 |
| MTN-014 | 14 days | Daily | Vagina | Rectum | 0 | 0 | 17,526 |
| MTN-014 | 14 days | Daily | Vagina | Vagina | 0 | 0 | 17,542 |
| MTN-017 | 56 days | Daily | Rectum | Rectum | 108 | 65 | 21,692 |
| MTN-017 | 56 days | Pericoital | Rectum | Rectum | 1 | 1 | 21,692 |

[a]Differentially expressed genes are defined by an FDR-adjusted $P$-value less than 0.05, with "up" indicating higher and "down" indicating lower expression during drug treatment relative to no treatment. "Total" indicates the number of genes that were detectable above background.

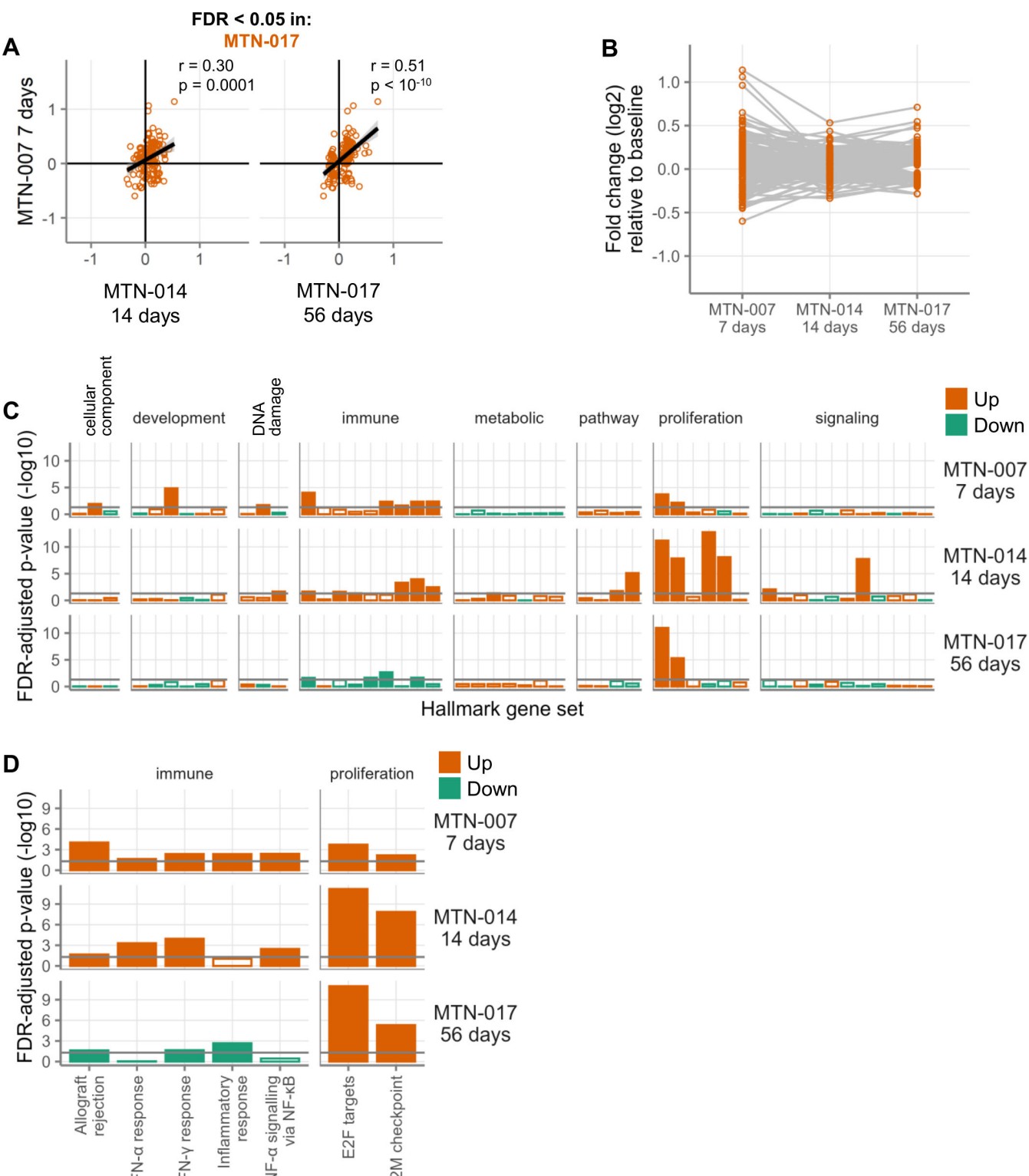

**FIG 2** Some gene expression changes persist in rectal tissue samples after daily topical tenofovir application to the rectum. (A) Correlation of gene expression changes between those genes that were differentially expressed after 56 days of gel use (MTN-017) and the same genes observed in MTN-014 and the prior study MTN-007. Each point represents the log2-fold change of a single gene from the studies indicated on the axes. Positive fold changes indicate higher expression during product use, and negative fold changes indicate higher expression at baseline. (B) The magnitude of fold changes of those genes that were differentially expressed after 56 days of gel use in MTN-017. (C) Gene set testing of the 50 Hallmark gene sets. Orange indicates gene sets containing genes that tended to be more highly expressed during treatment, and green indicates the opposite. Filled bars indicate FDR < 0.05. The gray horizontal line indicates FDR-adjusted $P$-value < 0.05. (D) The same plot as in panel C, but showing only those gene sets with FDR < 0.05 in at least 2 of the 3 study arms.

gene expression data, not just the differentially expressed genes. We did this by gene set testing using the 50 Hallmark gene sets (16), which represent a range of biological processes and features. As shown in Fig. 2C, a variety of gene sets were enriched in MTN-007 and MTN-014, with fewer gene sets enriched in MTN-017. Complete gene set results are available in File S2. Looking only at the gene sets that were significantly different in at least two study arms (Fig. 2D), we observed changes in gene sets related to cell proliferation and immunity. The proliferation gene sets were strongly and consistently upregulated at all time points and were the main gene sets affected after 56 days of use. The immune gene sets were upregulated after 7 and 14 days, but downregulated or unchanged after 56 days.

Similarly, when we performed gene set testing on the Gene Ontology (GO) Biological Process gene sets (17), the strongest signal was for enhanced cellular proliferation. Fifteen gene sets had FDR < 0.05 in at least two studies and showed the same direction in all three studies, almost all of which were related to cellular proliferation, especially gene sets that fell under the GO categories of DNA replication, DNA repair, mitotic cell cycle process, and protein localization to chromosomes (File S3). Taken together, these results suggest that tenofovir 1% gel induces expression of genes involved in cellular proliferation and/or DNA repair in the rectum and that this change endures for at least two months.

## Confirmation of rectal gene expression microarrays by RNA sequencing

To validate the results of our gene expression microarrays using another technique, we performed RNA sequencing (RNAseq) on the rectal tissue RNA from MTN-017. A total of 13,249 genes were detectable by both microarray and RNAseq. Of those, 10 were significantly differentially expressed at FDR < 0.05 by both techniques when comparing 56 days of daily use to baseline, 131 by microarray only, 25 by RNAseq alone, and the remaining 13,083 by neither (Fig. 3A). The fold changes were strongly correlated ($r$ = 0.64, $P$ < 1E-10) between microarray and RNAseq for the genes that were significant by microarray and/or RNAseq (Fig. 3A, left). Moreover, a correlation, albeit weaker, was also present for the genes that were not significant by microarray and/or RNAseq ($r$ = 0.35, $P$ < 1E-10) (Fig. 3A, right). The 10 genes that were significant by both techniques were all increased during tenofovir use and primarily related to cellular proliferation (such as the gene *POLQ* that encodes for DNA polymerase theta). Indeed, the top two hits by gene set testing of the RNAseq data were the same two gene sets as uncovered by the microarray data (E2F targets and G2M checkpoint), both of which are related to cellular proliferation (Fig. 2D, right). Thus, the RNAseq data essentially confirmed the results of the microarrays.

## Extension of gene expression changes to protein expression changes

We performed bottom-up proteomics using mass spectrometry on tryptic digests of rectal tissue samples and detected 1,785 proteins in MTN-014 and 1,565 proteins in MTN-017. Comparing baseline to treatment samples, no proteins were differentially expressed after adjustment by FDR in either study. Of the 1,565 proteins detectable by mass spectrometry in MTN-017, 12 of the corresponding RNA molecules were differentially expressed by microarray. The fold changes for these genes by microarray were strongly correlated with the fold changes for the corresponding proteins by mass spectrometry (Fig. 3B, $r$ = 0.85, $P$ = 0.0004). In contrast, there was little correlation between microarray and mass spectrometry fold changes for the genes that were not differentially expressed by microarray in MTN-017 (Fig. 3C; $r$ = 0.10, $P$ = 0.00006), and there was no such correlation in MTN-014 (Fig. 3D; $r$ = −0.007, $P$ = 0.77). This result indicates that significant differential expression at the RNA level likely translates to differential expression at the protein level, at least for those few proteins we were able to detect.

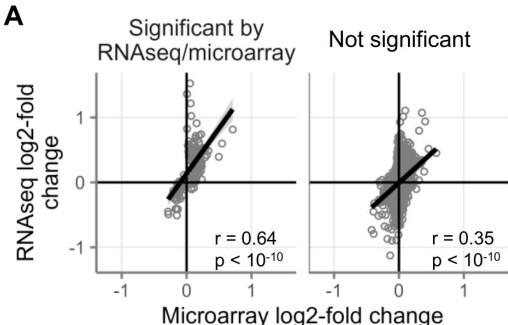

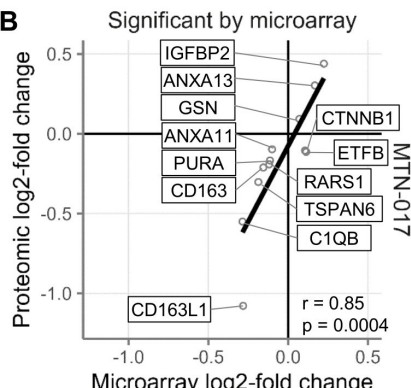

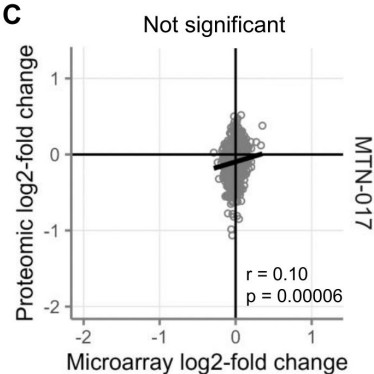

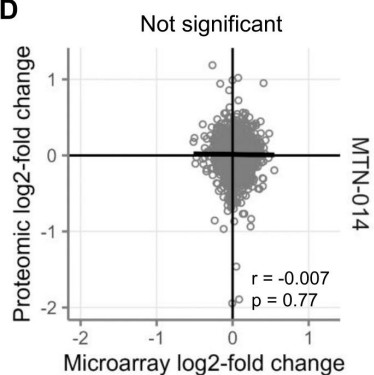

**FIG 3** Confirmation of gene expression changes by RNAseq and mass spectrometry. (A) Log2-fold changes comparing baseline to daily rectal tenofovir application in MTN-017. Plots show genes that were significant by FDR < 0.05 by at least one assay (left plot; *n* = 166 genes) or not significant by

**Fig 3 (Continued)**

either assay (right plot; $n$ = 13,083 genes). (B–D) Log2-fold changes comparing baseline to daily rectal tenofovir application measured by mass spectrometry (y-axis) and microarray (x-axis). (B) Fold changes for genes/proteins that were significant by FDR < 0.05 in the MTN-017 microarray. (C) Genes/proteins that were not significant in either assay in MTN-017. (D) Genes/proteins that were not significant in either assay in MTN-014.

## Pericoital use of tenofovir 1% gel

During one of the three study periods of MTN-017, participants used the topical gel pericoitally (before and after receptive anal intercourse) as an alternative to daily use. Gene and protein expression changes were well correlated between pericoital and daily use, when assessed by microarray (Fig. 4A), RNAseq (Fig. 4B), and mass spectrometry-based proteomics (Fig. 4C).

Despite the correlation of gene expression fold changes between daily and pericoital application, there were considerable differences between the two in terms of Hallmark gene set enrichment (Fig. 4D). As described above, the E2F targets and G2M checkpoint gene sets were highly enriched after daily application in all three studies. However, they were not significantly enriched after pericoital use. The downregulation of immune-related gene sets seen in the daily arm after 56 days (but not 7 or 14) was even stronger after pericoital use. In addition, several gene sets related to metabolism were upregulated after pericoital use; their direction of change was the same after daily use, but the changes after daily use were not significant by FDR < 0.05. Lastly, several gene sets were significantly changed during pericoital use that were not affected by daily use (Fig. 4D).

## Daily vaginal use of tenofovir 1% gel

In addition to rectal application, tenofovir 1% gel can be applied vaginally for protection during vaginal intercourse. It is unknown whether gene expression changes caused by tenofovir 1% gel in the rectum also occur in the vagina. The design of MTN-014 allowed us to assess the effect of daily vaginal tenofovir 1% gel application for 14 days on vaginal gene expression. As shown in Table 2, there were no differentially expressed genes. However, there was a surprisingly high correlation between rectal gene expression changes from MTN-007 and vaginal gene expression changes from MTN-014 (Fig. 5A), both for genes that were differentially expressed in the rectum in MTN-007 ($r$ = 0.19, $P$ < 1E-5 for genes upregulated in MTN-007 and $r$ = 0.06, $P$ = 0.18 for genes downregulated in MTN-007) and, to a lesser extent, for genes that were not ($r$ = 0.18, $P$ < 1E-10). This finding suggests that the gene expression changes seen in the rectum after 7 days occur to at least some extent in the vagina after 14 days of daily vaginal product application.

Despite the correlation of gene fold changes in the rectum and vagina, there were considerable differences between the two in terms of Hallmark gene set enrichment. Few of the gene sets affected in the rectal samples were affected in the vaginal samples and vice versa. In particular, the E2F targets and G2M checkpoint gene sets were not affected in the vagina. By gene set testing, the three Hallmark gene sets most strongly enriched in the vagina were related to the process of development: epithelial-mesenchymal transition, angiogenesis, and myogenesis (Fig. 5B).

## Effect of tenofovir 1% gel use across anatomical compartments

Finally, we sought to determine whether tenofovir's effect on gene expression carried over from one mucosal compartment to another, for example, whether gene expression changed in the rectum when the gel was used only in the vagina. We assessed gene expression in the rectum depending on whether tenofovir 1% gel had been applied to the rectum or the vagina, and we did the same in the vagina. There was a medium-strength correlation between the fold changes in the rectum when comparing rectal and vaginal application ($r$ = 0.49, $P$ < 1E-10, Fig. 6A). The correlation was stronger in the vagina when comparing rectal and vaginal application ($r$ = 0.64, $P$ < 1E-10, Fig. 6A).

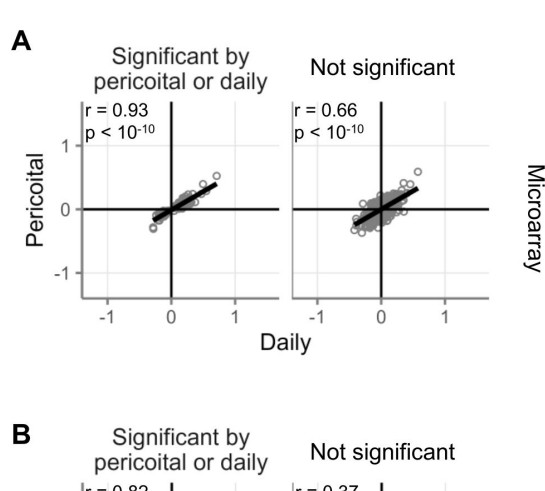

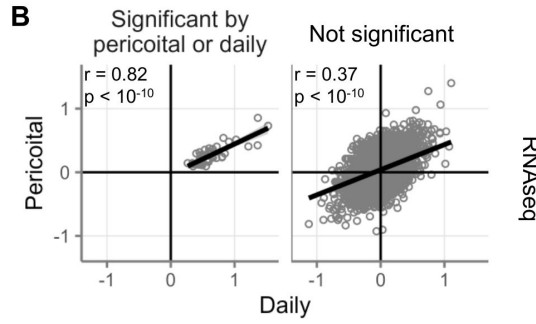

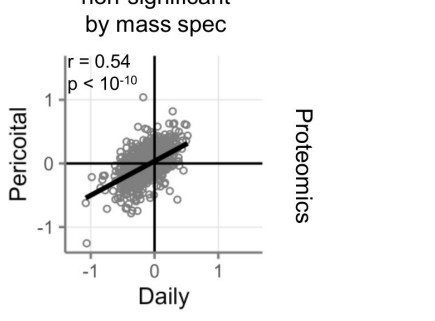

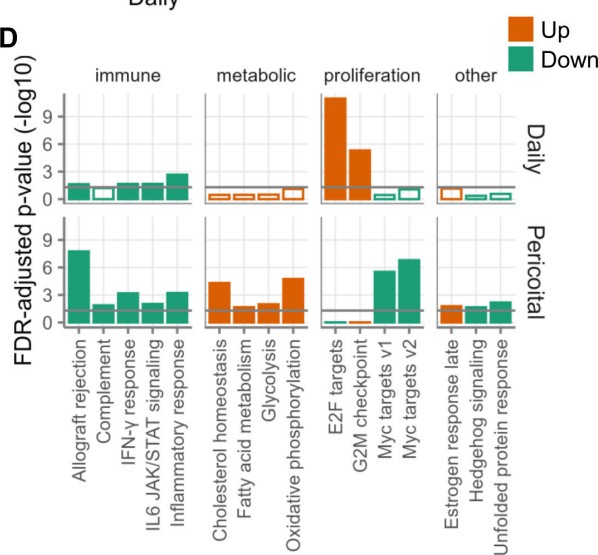

**FIG 4** Comparison of daily and pericoital application of topical tenofovir. Log2-fold changes comparing daily use of topical tenofovir to baseline are shown on the x-axis. Log2-fold changes comparing pericoital use of topical tenofovir to baseline are shown on the y-axis. Measurements were performed using

Fig 4 (Continued)

microarray (A), RNAseq (B), or proteomics (C). In panels A and B, the left column shows genes with FDR < 0.05 for daily ($n$ = 173 by microarray and $n$ = 48 by RNAseq) or pericoital ($n$ = 2 by microarray and $n$ = 0 by RNAseq) use. The right column shows genes with FDR ≥ 0.05. In panel C, all proteins had a non-significant FDR ≥ 0.05. (D) Gene set testing of Hallmark gene sets comparing daily and pericoital application of topical tenofovir (both in MTN-017 after 56 days of treatment). The only gene sets that are shown are those that had FDR < 0.05 for at least one of the two study arms. Orange indicates gene sets containing genes that tended to be more highly expressed during treatment, and green indicates the opposite. Filled bars indicate FDR < 0.05.

We also compared the magnitude of gene expression changes in the rectum or vagina between the two gel application routes. The distribution of fold changes in the rectum was wider with rectal application than with vaginal application (Fligner-Killeen test $P$ < 1E-10; Fig. 6B). The broader distribution of fold changes means that some genes have larger changes in rectal gene expression with rectal application than with vaginal application. Thus, rectal application is causing changes to gene expression, but it is so subtle that we are not able to detect it at the individual gene level (Table 2). In contrast, the distribution of fold changes was very similar in the vagina whether tenofovir was applied rectally or vaginally, although the difference was still statistically significant (Fligner-Killeen test $P$ = 0.002; Fig. 6B). Taken together, these results indicate that tenofovir has a smaller effect on rectal gene expression when applied vaginally than when applied rectally, but has an equal effect on vaginal gene expression regardless of the site of application.

When looking at Hallmark gene sets in the rectum, we observed broader and stronger changes in the rectum after rectal topical application of tenofovir than after vaginal application (Fig. 6C), although the direction of change generally remained the same. When looking at Hallmark gene sets in the vagina (Fig. 6D), changes appeared more similar between rectal and vaginal tenofovir 1% gel application, particularly for gene sets related to development.

Interestingly, in both the rectum and the vagina, several gene sets related to the immune system were upregulated after rectal application of tenofovir, but not after vaginal application. This suggests that rectal application may be more inflammatory than vaginal application and that this inflammatory signal may spread to the vagina.

## Summary of correlations between studies

Our main findings are the correlations between every arm of the new studies (MTN-014, MTN-017) with MTN-007 (daily rectal application for 7 days in rectal tissue samples), which are shown in Fig. 1, 2 and 5. To summarize those findings and put them into context, we re-plotted each correlation coefficient in Fig. 7. We separately show the correlations for all genes and for only those genes that were differentially expressed in MTN-007 (showing up and down separately).

The summary figure shows that for the effects of daily rectal application, the overall correlation with MTN-007 is higher after 14 days than after 56 days; the same applies to significantly downregulated genes. For significantly upregulated genes, correlations with MTN-007 were roughly similar after 14 and 56 days. Thus, the rectal gene expression changes seen after 7 days of daily rectal application wane after 14 and 56 days.

The effects of daily vaginal application on vaginal tissue had a high overall correlation with the data from the rectum in MTN-007, indicating that similar gene expression changes are induced in vaginal tissue as in rectal tissue. In fact, even daily rectal application induced similar changes in vaginal tissue. In contrast, daily vaginal application had a smaller effect on gene expression in rectal tissue. Overall, this suggests higher absorption of tenofovir from the rectum and transfer to the vagina than *vice versa*.

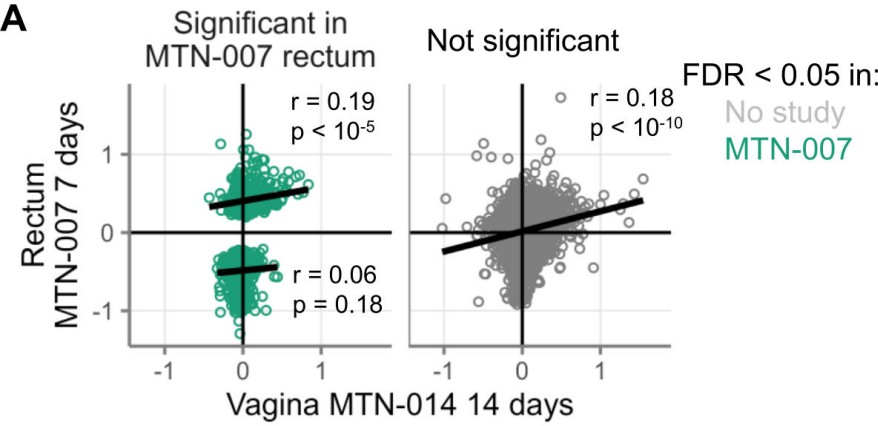

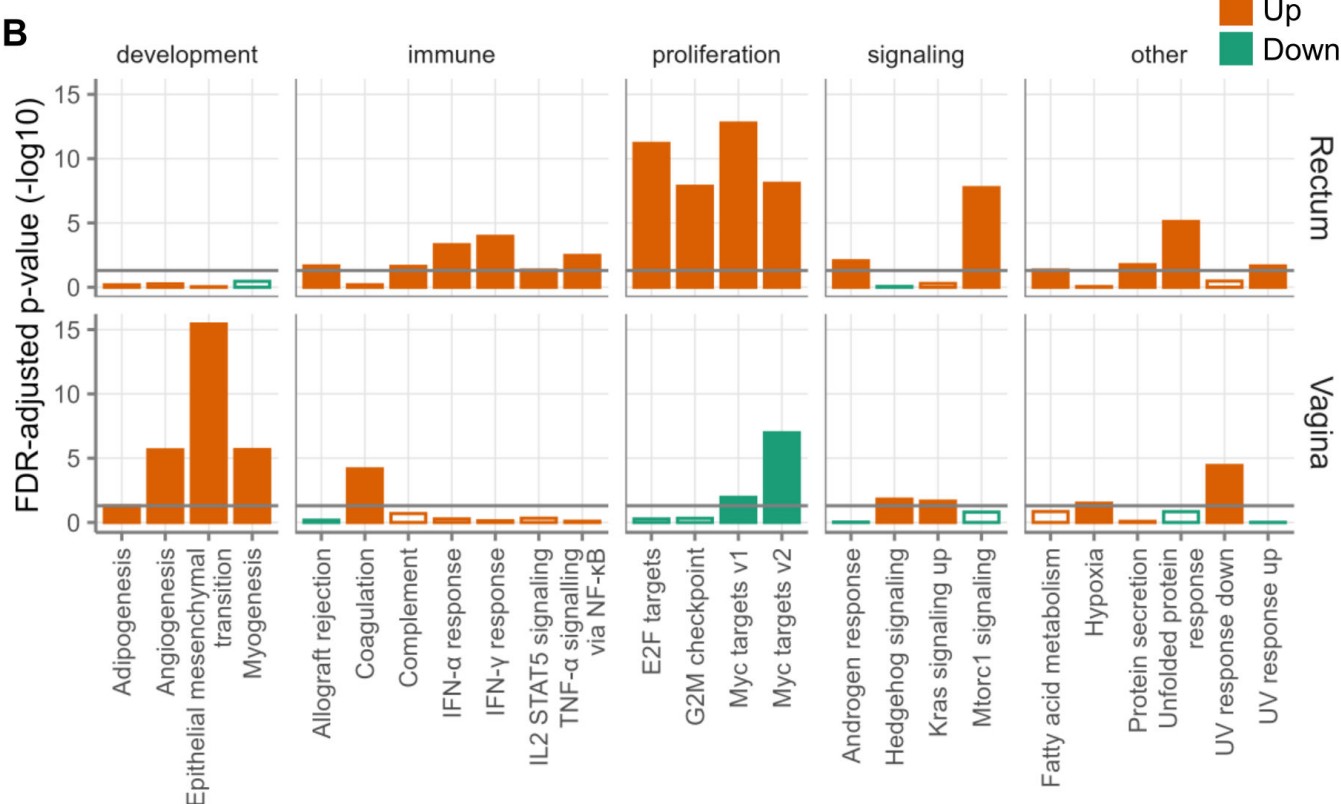

FIG 5  Gene expression changes in vaginal tissue samples after daily topical tenofovir application to the vagina. Biopsies were taken from the vagina after 14 days of daily vaginal gel application (MTN-014). (A) Each point represents the log2-fold changes of a single gene (averaged across multiple probes and subjects as described in the Methods) from the two studies indicated on the axes. Positive fold changes indicate higher expression during product use, and negative fold changes indicate higher expression at baseline. (B) Hallmark gene sets with FDR < 0.05 after 14 days of daily gel application in MTN-014. The top row shows gene set changes in the rectum after rectal application. The bottom row shows gene set changes in the vagina after vaginal application. The only gene sets that are shown are those that had FDR < 0.05 for at least one of the two study arms. Orange indicates gene sets containing genes that tended to be more highly expressed during treatment, and green indicates the opposite. Filled bars indicate FDR < 0.05. The gray horizontal line indicates FDR-adjusted P-value < 0.05

## Comparison of topical tenofovir effects to those of oral TDF/FTC use

While this article focuses on topical application of tenofovir, this drug is also commonly used in oral form. In a previous study (7), we showed that daily oral TDF/FTC (Truvada) as pre-exposure prophylaxis (PrEP) caused significant upregulation of certain type I/III interferon-stimulated genes (ISGs) in the rectum, including IFI6, ISG15, and MX1. Therefore, we sought to assess whether topical use of tenofovir in the rectum

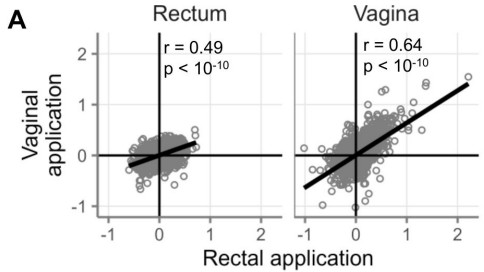

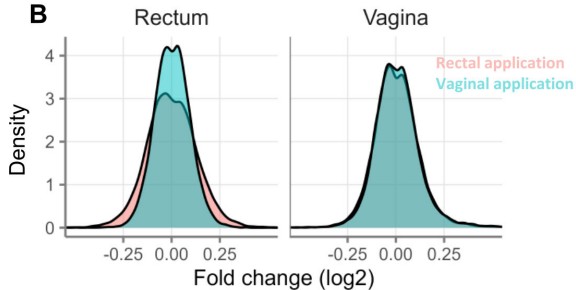

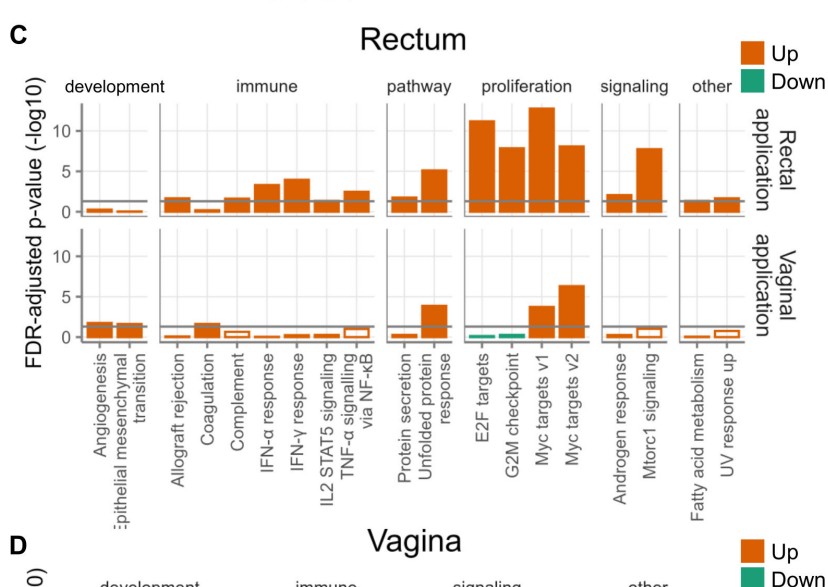

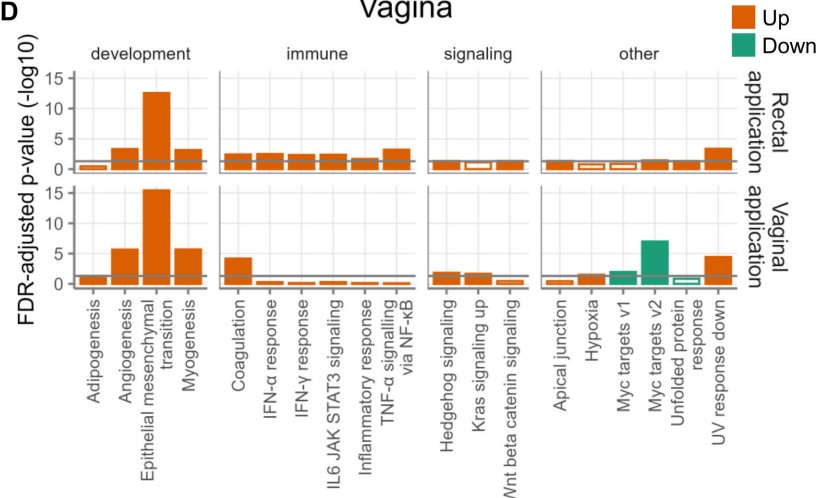

**FIG 6** Gene expression changes after cross-compartment application. All samples come from MTN-014. (A) Gene expression changes in the rectum (left) or vagina (right) after topical application of tenofovir to the rectum (x-axis) or vagina (y-axis). Each point represents the log2-fold change of a single gene (Continued on next page)

Fig 6 (Continued)

resulting from the application routes indicated on the axes. (B) Distribution of fold changes in the rectum (left plot) or vagina (right plot) after rectal (pink) or vaginal (teal) application of tenofovir. The x-axis is cut off at ±0.5 to emphasize the central portion of the fold changes. (C) Hallmark gene sets with FDR < 0.05 in biopsies of the rectum after 14 days of rectal or vaginal application. The only gene sets that are shown are those that had FDR < 0.05 for at least one of the two study arms. Orange indicates gene sets containing genes that tended to be more highly expressed during treatment, and green indicates the opposite. Filled bars indicate FDR < 0.05. (D) Hallmark gene sets with FDR < 0.05 in biopsies of the vagina after 14 days of rectal or vaginal application.

upregulates these same genes. As shown in Fig. 8A, there were strong correlations for the fold changes in the rectum of these genes between oral TDF/FTC PrEP (tested as a third study arm in MTN-017) and topical use of tenofovir in MTN-007, MTN-014, and MTN-017. This indicates that the same upregulation of these type I/III interferon-stimulated genes caused by oral use of TDF/FTC is also caused by topical use. Unlike the overall waning of effect size for most genes, we observed that the fold changes of ISGs diminished slightly from 7 to 14 days, but then trended to increase again between 14 and 56 days (Fig. 8B).

As an alternative method to compare the changes induced by oral use of TDF/FTC to the changes induced by topical use of tenofovir, we used two methods of gene set enrichment testing. First, we took the set of genes that were differentially expressed after oral use ($n$ = 13 genes) and tested them for enrichment in the rectal application data. This set of genes, all of which were upregulated in the oral arm of MTN-017, was significantly enriched in the same direction in the daily topical tenofovir arms of MTN-007 (FDR = 2.6E-4), MTN-014 (rectal arm; FDR = 9.9E-3), and MTN-017 (FDR = 1.8E-15). Notably, the same gene set was also significantly enriched in the same direction in the vagina after 14 days of daily topical tenofovir application to the vagina (vaginal arm of MTN-014; FDR = 3.9E-2). These results complement the correlation results.

Second, we tested an expanded list of 24 type I/III ISGs compiled from the results of our prior oral TDF/FTC studies (7) and used these as a single gene set ("ISG-24" gene

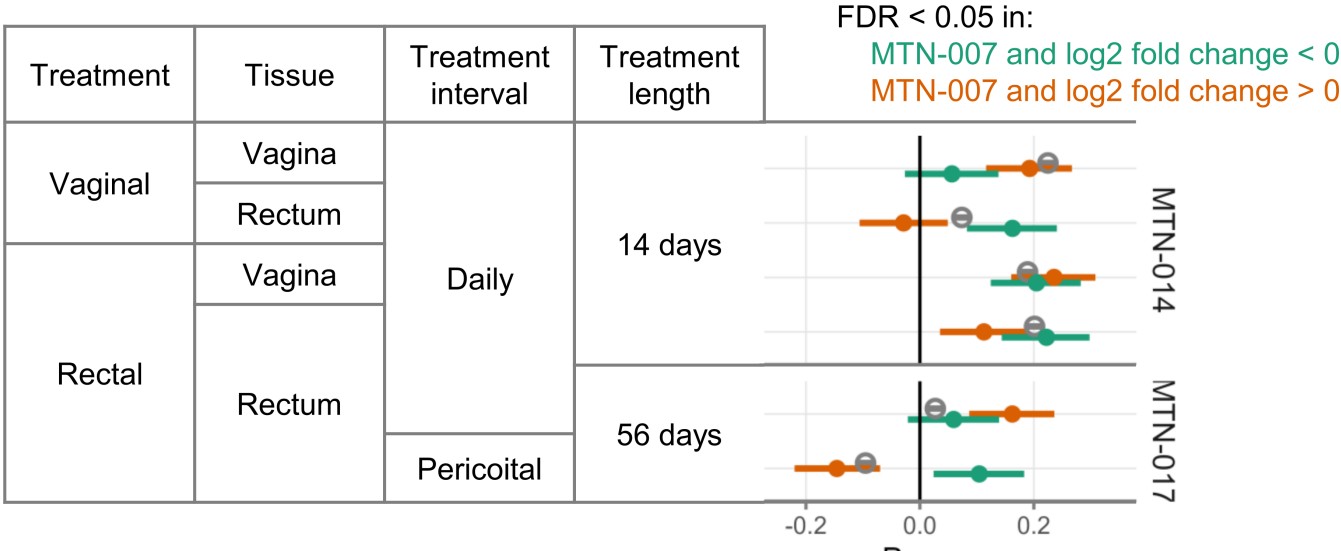

FIG 7 Summary of correlations with MTN-007. Correlations of gene expression fold changes between MTN-007 (7 days of daily rectal 1% tenofovir gel application) and the gene expression fold changes from the two new studies presented here. Positive Pearson $r$ values indicate positive correlations between the fold changes from MTN-007 and the fold changes from the various study arms in MTN-014 and MTN-017. The open gray symbols indicate correlations calculated on all genes that were not differentially expressed in MTN-007. The filled green and orange symbols indicate correlations calculated on genes that were differentially expressed in MTN-007 (green for genes with significant negative fold changes and orange for genes with significant positive fold changes in MTN-007). The error bars indicate the 95% confidence intervals for the correlation coefficients.

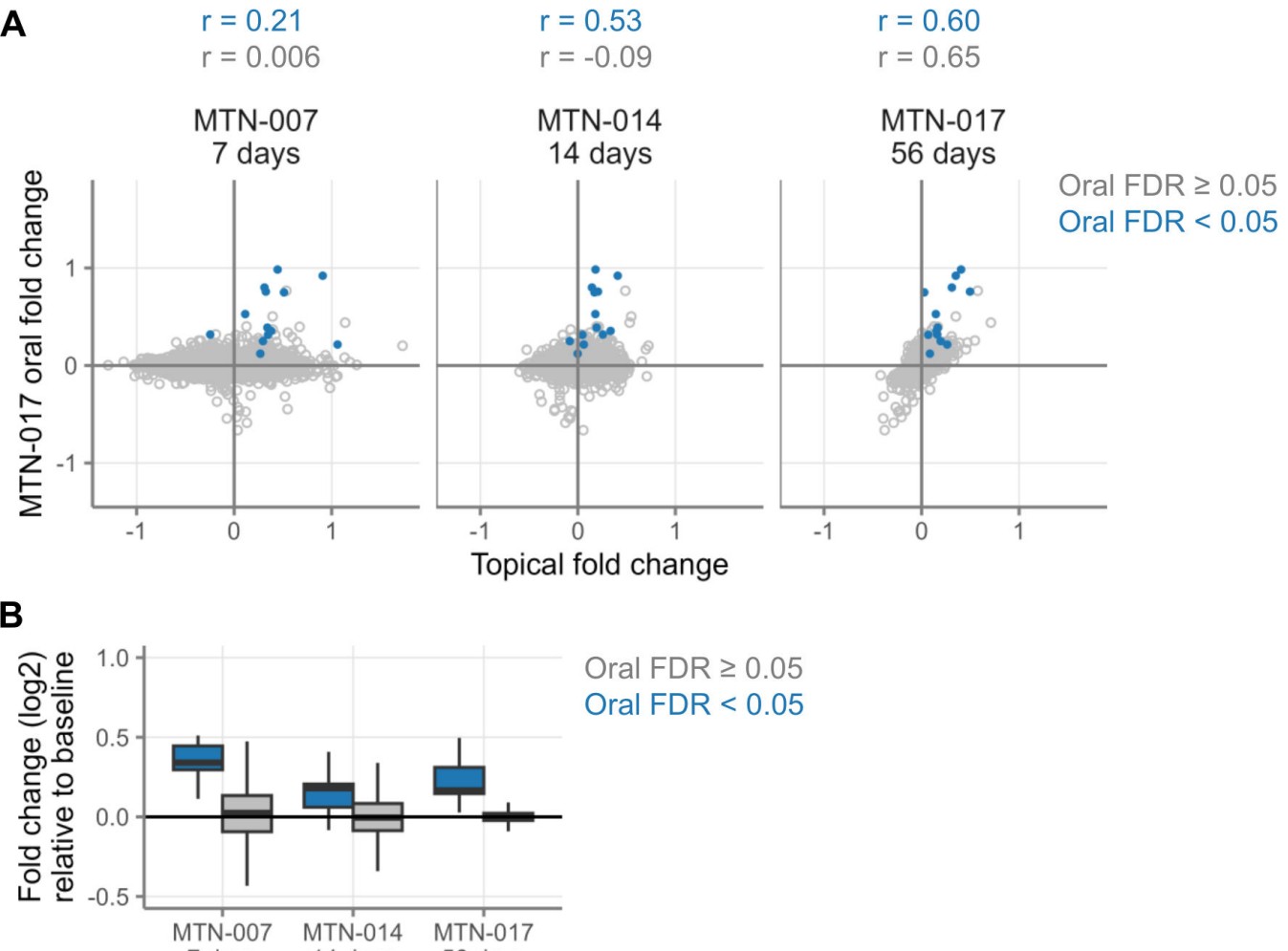

**FIG 8** Comparison of topical and oral products on gene expression in the gut. (A) Each point represents the log2-fold changes of a single gene from the studies indicated. Positive fold changes indicate higher expression during product use, and negative fold changes indicate higher expression at baseline. Blue color indicates that the gene had an FDR-adjusted $P < 0.05$ in the oral arm of MTN-017 ($n$ = 13 genes). Twelve of these 13 genes are associated with type I/III interferon pathways. (B) Magnitude of fold change over time of genes after topical use for 7, 14, or 56 days, with blue showing the genes that were differentially expressed in the oral arm and gray showing all other genes.

set). This list included all ISGs with log2-fold change > 0.25 in the rectum or duodenum after 56 days of oral TDF/FTC, either by microarray or RNAseq, independent of FDR (Table S1). Given this low fold change cutoff, this approach is inclusive and may include false positives but is designed to capture as many ISGs potentially affected by oral TDF/FTC as possible. The ISG-24 gene set was significantly enriched in the positive direction in the daily topical tenofovir arms of MTN-007 (FDR = 7.4E-5), MTN-014 (FDR = 2.0E-4), and MTN-017 (FDR = 2.3E-7). Notably, the significance of ISG-24 enrichment increased with longer use. Of note, it was even enriched in the MTN-017 arm of only pericoital tenofovir 1% gel use (FDR = 4.1E-4). Lastly, ISG-24 was also enriched in the vagina after 14 days of daily vaginal (FDR = 9.9E-3) or daily rectal (FDR = 0.032) tenofovir use. Thus, together, the results of our current paper and the results of our prior papers (5, 7) demonstrate a consistent stimulatory effect of tenofovir on type I/III ISG expression over various times of topical 1% gel use, upon oral use in the form of TDF/FTC PrEP, and at different anatomical locations (rectum, duodenum, and vagina).

## DISCUSSION

One topical HIV preventative has been approved for distribution in some countries: a vaginal ring releasing dapivirine, a non-nucleoside reverse transcriptase inhibitor (11–14). The nucleotide reverse transcriptase inhibitor (NRTI) tenofovir has also been extensively tested as a topical preventative against both vaginal and rectal HIV transmission, but so far has not reached the licensing stage (18–22). Studying the effects of topical tenofovir on the mucosa is worthwhile for two reasons. First, it provides information about what kind of side effects could be expected should a topical product become available. Second, it may help to gauge how oral TDF as part of antiretroviral combination therapy in people living with HIV (PWH) may alter the mucosal environment, especially in the gut.

Here, we assessed topical tenofovir's effect on the gut and vaginal mucosa over 7, 14, and 56 days in three microbicide trials (MTN-007, MTN-014, and MTN-017) and compared our findings to oral use of TDF/FTC. We find that most gene expression changes diminish over longer periods of gel use, but that stimulation of the type I/III interferon system and of cell proliferation persists. We found a few differentially expressed genes in most study arms, which is reassuring. Given the varying sample sizes between the studies, we did not strictly filter for differentially expressed genes, but additionally assessed similarities in fold changes between studies.

### Both topical and oral formulations of tenofovir stimulate the type I/III interferon system in the mucosa, even in the absence of HIV

Our key finding is the same inducing effect on type I/III interferon-related (IFN I/III) genes as we found in our prior study of oral TDF/FTC (tenofovir/emtricitabine) (7). While the stimulatory effect of topical tenofovir on the rectal IFN I/III system appeared milder than that of oral TDF/FTC, it did not wane over time (Fig. 8B). Taken together, our studies therefore show that tenofovir, and perhaps other NRTIs, stimulate the interferon system in the mucosa. Whether this is beneficial or detrimental remains to be determined. On the one hand, activation of the IFN I/III system could contribute to antiviral efficacy. On the other hand, if this stimulation persists over longer periods of drug use, it could amount to a smoldering interferonopathy and contribute to the increased chronic disease burden in PWH, including accelerated aging ("inflammaging") (23–26). This as-yet speculative concern should be further studied.

Several mechanisms could explain how tenofovir stimulates the type I/III IFN system. First, NRTI drugs were shown to increase the proliferation rate of epithelial cells (27, 28). In our prior study of oral pre-exposure prophylaxis (PrEP), daily TDF/FTC induced proliferation, especially on a distinct subpopulation of enterocytes expressing high levels of interferon-stimulated genes (ISGs) (7). More of these ISG^high enterocytes should translate into higher activity of the interferon system in the gut. Second, Murata et al. have shown that tenofovir reduces anti-inflammatory interleukin 10 (IL-10) signaling. Mechanistically, they showed that tenofovir monophosphate, an intracellular metabolite of TDF, strongly binds to Akt, a protein kinase involved in the intracellular signaling cascade necessary for lipopolysaccharide (LPS)-induced IL-10 transcription (29). This binding prevents Akt phosphorylation, interrupting a key event in the LPS/IL-10 signaling cascade. Consequently, TDF treatment reduces anti-inflammatory IL-10 responses to LPS (as we also previously showed with topical tenofovir) (5) in favor of a pro-inflammatory environment. This would likely include ISG induction, although the authors did not specifically study the IFN system in their model. Finally, NRTIs can cause the accumulation of RNA:DNA hybrids that activate the innate immune system. Rajurkar et al. reported that the NRTI lamivudine (3TC) led to the accumulation of RNA:DNA hybrids derived from aborted reverse transcription of repeat RNA species derived from long interspersed nuclear element-1 (LINE-1) retrotransposons and human endogenous retroviruses (HERV) (30). These RNA:DNA hybrids are sensed by the cell-intrinsic innate system, for example, by cGas (31), TLR9 (32), and perhaps a number of other cytosolic

proteins (33), leading to ISG induction. This same mechanism could be postulated for tenofovir. Lastly, it is well known that tenofovir causes some mitochondrial toxicity, which in liver cells has been linked to activation of the pro-inflammatory NF-κB signaling pathway via elevated mitochondrial reactive oxygen species (34). Taken together, there are several plausible explanations for how tenofovir leads to activation of the IFN system, but more studies are needed to understand this link completely.

## Except for type I/III interferon and cellular proliferation pathways, gene expression changes generally waned with longer topical tenofovir use

Genes belonging to processes involved in the regulation and induction of cell proliferation were also upregulated across all three time periods tested (Fig. 2D). In contrast, the majority of gene expression changes observed after 7 days of daily topical tenofovir were reduced after 14 days of use and decreased even further over 56 days of daily use (Fig. 1C). Thus, topical tenofovir 1% gel applied daily to the rectal mucosa induced genes signifying activation of (i) type I/III interferon and (ii) cellular proliferation pathways that endured for at least two months. All other gene expression changes appeared to wane over time. These findings would suggest that the prolonged use of tenofovir 1% gel is likely tolerable to the mucosa, consistent with participant reports (8, 9). However, potential side effects, such as mucosal inflammation related to chronic stimulation of the type I/III IFN system and/or epithelial hyperplasia related to increased cellular proliferation, should be monitored. The need for monitoring is also highlighted by a study reporting vaginal ulcer formation in 8/12 participants one month after placement of a vaginal TDF ring (21). Lastly, whether our findings here for topical tenofovir PrEP, or those reported previously for oral TDF/FTC PrEP (7), warrant consideration of NRTI drugs as potential contributors to residual chronic inflammation and the rise of non-AIDS-defining cancers in PWH remains an open question.

Interestingly, the strong induction of gene sets related to cell proliferation that we saw after topical use was not observed in the rectum after oral TDF/FTC use in MTN-017 (7). This result was mirrored in rectal tissue from another trial (ACTU-3500) of eight participants using oral TDF/FTC (7). However, cell proliferation gene sets were induced in duodenal tissue in that study. Drug concentrations likely reach much higher levels in the duodenum than in the rectum after oral administration. Further, local tissue concentrations are much higher with topical administration of the gel than with oral use (35–37). Thus, high drug concentrations, such as those achieved with topical administration in the rectum and vagina, or in the duodenum after oral use, may be necessary for the induction of cell proliferation-related gene sets. In contrast, the induction of type I/III interferon-related genes may occur with lower drug levels, as this phenomenon was observed in all study arms. Additionally, some effects observed with topical gel use but not oral use may result from the hyperosmolarity of the gel itself. While all studies here used the reduced-glycerin gel formulation, which is less hyperosmolar than previous formulations, it remains hyperosmolar, which may contribute to changes in gene expression.

## Intermittent pericoital use of rectal tenofovir 1% gel caused fewer but some unique gene expression changes

An alternative to the daily use of tenofovir 1% gel could be application only before and after anticipated sexual intercourse, which might induce fewer changes in the mucosa and therefore be safer. In MTN-017, we were able to assess gene expression changes over 56 days of such pericoital tenofovir 1% gel use. As previously reported, participants used the gel an average of 2.9 times per week during the pericoital regimen and in total approximately half as often as during the daily use study phase (15). While the entirety of gene expression changes was highly correlated between daily and pericoital use (Fig. 4A and B), gene sets related to cellular proliferation did not increase during pericoital use (Fig. 4D). This provides some evidence that pericoital use could indeed be safer than daily use. Interestingly, however, pericoital use did induce genes related to metabolic

pathways, including those involved in cholesterol homeostasis and fatty acid metabolism (Fig. 4D). A recent study found that oral TDF/FTC PrEP users exhibited higher serum levels of intestinal fatty acid-binding protein (I-FABP), suggesting that TDF/FTC may also influence the metabolism and transport of fatty acids in the intestinal mucosa (38). The fact that this finding was associated with faster clearance of an intravenously infused protective anti-HIV monoclonal antibody (38, 39) additionally highlights that topical or oral PrEP may influence the efficacy of other concurrently delivered HIV prevention strategies.

## Findings from the vaginal/rectal cross-compartmental study (MTN-014)

The unique crossover design of MTN-014 allowed us to directly compare the effects of tenofovir 1% gel between the rectal and vaginal mucosa within the same study participants. While we found a strong correlation between the rectal gene expression changes in MTN-007 after 7 days and the vaginal gene expression changes in MTN-014 after 14 days (Fig. 5A), this correlation did not hold when comparing rectal and vaginal gene expression at 14 days within MTN-014. Correspondingly, we also found very different results at the gene set level. Most notably, tenofovir application to the vagina induced the vaginal mucosa to upregulate gene sets related to epithelial-mesenchymal transition and angiogenesis (Fig. 5B). Epithelial-mesenchymal transition (EMT) is the process by which epithelial cells acquire motile and invasive characteristics consistent with mesenchymal cells. EMT plays a critical role during early embryonic development, but during later stages in life, it can also be involved in the generation of cancer stem cells and resultant malignant disease processes (40). While no publications exist specifically for topical tenofovir, vaginal treatment with the NRTI azidothymidine in mice yielded a genital cancer rate of ~25% (27). Thus, the prominent induction of an EMT-related gene set in the vaginal mucosa by tenofovir 1% gel could be a concern if it persisted over longer periods of use.

In MTN-014, tissue samples were taken from both the vaginal and rectal mucosa at screening and after each product-use period. We could therefore also assess whether gel application to one mucosal compartment affects the other compartment. This cross-compartmental comparison revealed that, unsurprisingly, tenofovir had a smaller effect on rectal gene expression when applied vaginally than when applied rectally. However, it had an equal effect on vaginal gene expression regardless of the site of application (Fig. 6A and B). This included, for example, strong induction of the EMT gene set in the vagina not only after vaginal but also after rectal administration (Fig. 6D). Likewise, type I/III and type II interferon pathways were induced in the vagina after rectal gel use (Fig. 6D).

Studies in non-human primates and humans have assessed the distribution of tenofovir across these two compartments (8, 36, 41). Yang KH et al. were the first to demonstrate in humans the presence of tenofovir in vaginal fluid after rectal tenofovir 1% gel administration (36). In MTN-014, the cross-compartmental drug measurements after rectal dosing found ~1/100th of the rectal fluid tenofovir concentrations in vaginal fluid (8). In the opposite direction, after vaginal dosing, MTN-014 found ~1/1,000th of the vaginal fluid tenofovir concentrations in rectal fluid. However, the smaller drug gradient from the rectum to the vagina than *vice versa* likely does not explain the vagina's equal reactivity to rectal or vaginal TFV application in our gene expression analysis, since the gradient was still large. Rather, more complex biological causes may be at play. For example, the vaginal mucosa could be inherently more sensitive to TFV than the rectal mucosa. Alternatively, at least for epithelial cell-specific pathways such as epithelial-mesenchymal transition, the multiple cell layers of the squamous vaginal epithelium provide many more responder cells than the single-layer columnar epithelium of the rectum. Either scenario could explain our ability to detect a vaginal response across a presumably wide range of drug concentrations. Notably, while we detected expression changes at the gene set level in the vagina, no genes were differentially expressed, indicating that the changes are small; the similarity in responses in the vagina with vaginal and rectal application may simply indicate that gel application has little effect on vaginal gene

expression. Lastly, drug efficacy may also be influenced by differing TFV-metabolizing activities of the physiological vaginal and rectal microbiomes. Gram-negative bacteria, which tend to be absent from the vagina but colonize the rectum, have been reported to metabolize and partially inactivate vaginally administered tenofovir (42).

## Proteomic studies

Our unbiased proteomic analysis using mass spectrometry provided complementary data to our transcriptomics data. However, we could not confirm or disprove the main findings of the gene expression studies because too few proteins were detectable in the previously described cell proliferation and interferon gene sets. This may have been due to the limited amount of tissue available for the protein study and/or the inherently lower detection sensitivity of mass spectrometry compared to RNA sequencing. Notably, however, for differentially expressed genes where the corresponding proteins were detectable, the correlation between gene and protein expression changes was very strong (Fig. 3B). Among these, the downregulation of CD163 and CD163L1 at the gene and protein levels in MTN-017 deserves mention. Both molecules are expressed by macrophages, with CD163L1 marking those that produce IL-10. A counterpoint to CD163L1 is CLEC5A, a C-type lectin characteristic of macrophages that produce TNFα (43). CLEC5A was not detected by mass spectrometry, but its gene expression was significantly upregulated. The combination of CD163L1 downregulation and CLEC5A upregulation suggests a pattern of increased inflammatory, TNFα-producing, and fewer anti-inflammatory, IL10-producing macrophages, induced by daily rectal tenofovir 1% gel application over 56 days. This finding is consistent with published *in vitro* modeling of tenofovir's effect in lipopolysaccharide-stimulated peripheral blood-derived monocytes (29), as well as our own prior *in vivo* and *in vitro* findings (5).

## Impact of biological sex on responses to tenofovir 1% gel

All participants in MTN-007 and MTN-017 were of presumed male biological sex, whereas all participants in MTN-014 were of presumed female biological sex. Immune responses have been reported in many studies to differ between biological sex (44). We are unable to address this question directly because each study was processed as a separate batch, so any comparisons between biological sex, such as of baseline samples, would be seriously confounded by batch effects. However, the changes in rectal gene expression due to application of tenofovir 1% gel were largely congruent between studies, indicating that many of tenofovir's effects are likely similar across biological sex. Lastly, it remains possible that the lack of differentially expressed genes seen in MTN-014 is related to the fact that all participants in that study were female, unlike MTN-007 and 017.

## Limitations

A limitation of our study was the comparison of gene expression changes between three separate clinical trials, with different individuals recruited into each protocol and different sample sizes in each study. Nevertheless, gene expression changes within each study were always determined in comparison to a within-participant baseline without drug, providing a much more reliable measurement of biological effect than would be possible cross-sectionally. This provides high confidence that whatever changes were uncovered likely were a consequence of tenofovir treatment and not something else. However, we were not able to determine whether individuals of different biological sex, age, ethnicity, or other demographic variables responded differently to tenofovir. In addition, differential adherence to gel use between studies could have affected our results. In MTN-007, all participants reported using the gel at least 80% of the time (6). In MTN-014, 91% of doses were directly observed by study staff (8). In MTN-017, 83% of participants reported using the gel at least 80% of the time during the daily rectal study period (15). Thus, adherence was high across all studies, but some differences could have

caused us to underestimate the effect of gel use after 56 days, because adherence did appear to decrease with the time of gel use. For example, several Hallmark immune gene sets were upregulated after 7 and 14 days, but not after 56 days (Fig. 2C and D). Furthermore, our studies were conducted in the absence of HIV infection, so our findings are most relevant to tenofovir's use as a preventative agent rather than a treatment.

In summary, across three separate tenofovir-based HIV pre-exposure prophylaxis trials, induction of type I/III interferon-related genes was the most consistent and persistent mucosal response, occurring after both oral and topical use and at all tested time points. Increased type I/III IFN signaling could enhance the antiviral preventive efficacy of tenofovir. However, over long periods of use, persistent stimulation of IFN pathways could also have detrimental effects, such as contributing to chronic immune activation and its associated co-morbidities in PWH. We believe that this clinical concern should be further investigated. Still, we also emphasize that tenofovir and other NRTIs will continue to be essential components of ART, offering huge benefits over leaving HIV infection untreated or ineffectively treated.

## MATERIALS AND METHODS

### Studies

Samples were used from two new studies, which are described in Table 1. The Microbicide Trials Network trial 014 (MTN-014) was a crossover study of reduced-glycerin tenofovir 1% gel applied topically to the rectum or vagina for 14 days, with a six-week washout between compartments (8). For this study, rectal and vaginal tissue samples were available from 12 participants at three time points: baseline, the end of phase I (rectal or vaginal application), and the end of phase II (rectal or vaginal application). The Microbicide Trials Network trial 017 (MTN-017) included oral TDF/FTC as well as rectal reduced-glycerin tenofovir 1% gel (9). The data on oral TDF/FTC has been previously published (7); the topical samples are the focus of this analysis. Rectal tissue samples were available from 36 participants at four time points for: at baseline, after phase I, after phase II, and after phase III, where each phase represents two months of daily topical gel use, pericoital topical gel use, or daily oral TDF/FTC use. Participants were randomized to treatment order. MTN-007 was reviewed and approved by the Institutional Review Board (IRBs) of the University of Pittsburgh, University of Alabama, and Fenway Health. MTN-014 was reviewed and approved by the Columbia University IRB. MTN-017 was reviewed and approved by the IRBs and ethics committees of all eight participating clinical sites (four in the United States, two in Thailand, one in Peru, and one in Cape Town). All subjects in all three studies provided written informed consent. Complete sample size information and ClinicalTrials.gov identifiers are listed in Table 1. In MTN-007, rectal tissue samples were obtained at 9 and 15 cm from the anus. In MTN-014 and MTN-017, all rectal tissue samples were obtained at 15 cm from the anus. In the discussion of MTN-007 in this manuscript, we use only the tissue samples from 15 cm for consistency with the other studies.

### Adherence

Dosing was directly observed in MTN-014, with 91% of doses observed by study staff. As reported in the primary manuscript for MTN-014, tenofovir was detected in 92% of vaginal tissue samples after vaginal application and in 86% of rectal tissue samples after rectal application (8). In MTN-017, 93% of participants adhered to pericoital use at least 80% of the time and 83% adhered to daily use at least 80% of the time (15).

### Sample processing

Vaginal and rectal tissue samples were obtained in MTN-014 and MTN-017 as described in the primary manuscripts (8, 9). Tissue samples were transferred to RNALater

Stabilization Solution (ThermoFisher Scientific, Waltham, MA, USA), refrigerated at 4°C for 24 h, and finally frozen at −80°C. RNA was extracted as described previously (7). Briefly, tissue samples were homogenized, and RNA was extracted using the RNeasy fibrous tissue mini kit (Qiagen). For quality control, RNA integrity numbers were measured using TapeStation R6K assay (Agilent, Santa Clara, CA, USA).

## Microarray labeling and hybridization

Samples were labeled for microarray using the Illumina TotalPrep RNA Amplification kit (ThermoFisher) with 50 ng of RNA from MTN-014 and 500 ng from MTN-017, based on available RNA. A total of 750 ng of labeled cRNA from each sample was used for hybridization with HumanHT-12 v4 Expression BeadChips (Illumina, San Diego, CA, USA) at the Fred Hutch Genomics Core Facility.

## RNA sequencing

RNA sequencing was performed as described previously (7). In brief, libraries were prepared using the TruSeq Stranded Total RNA with Ribo-Zero Globin kit (Illumina) with 300 ng input. Libraries were sequenced on a HiSeq 2500 (Illumina) instrument (Illumina) with a 50 base pair, paired-end run at 50 million reads per sample.

## Mass spectrometry proteomics

Proteomics was performed as described previously (7). Briefly, biopsies were lysed, homogenized, and filtered prior to digestion using 2 µg trypsin (Promega, USA) per 100 µg of protein. Following digestion, 1 µg of peptide per sample was analyzed by LC-MS/MS using a Velos Orbitrap (Thermo Fisher) tandem mass spectrometer.

## Statistical analysis

Microarrays were analyzed as described previously (5, 7). Briefly, microarrays were pre-processed separately for each study and sample type. Variance stabilizing transformation (45) was applied, followed by robust spline normalization from the lumi R package (46). Probes that were rarely expressed (detectable in fewer than 10 samples per data set) were removed. In addition, several probes were found to differ between treatment groups at baseline in MTN-007. These probes were removed before further analysis. During quality control assessment, two pairs of vaginal and rectal samples in MTN-014 were determined by gene expression profiles to have been switched. In both cases, the pairs were physically adjacent during sample processing, and the sample labeled "rectum" had gene expression consistent with vagina and vice versa. The samples were switched back before further analysis.

Differential gene expression was determined using the limma package (47). For each sample type and study, we fit paired models, comparing baseline to treatment samples within each participant. Differential expression was defined as any probe having an adjusted $P$-value (FDR) less than 0.05. We performed gene set testing with camera from limma (48). Because there are multiple probes for some genes, probes were collapsed into genes by taking the probe with the lowest adjusted $P$-value (FDR) for each gene (49).

Pearson's product-moment correlation coefficient was used to assess correlations between different arms of studies. We assessed the distribution of fold changes and tested for differences in the variances using the Fligner-Killeen test, a non-parametric test used to compare variances between groups. $P$-values below 1E-10 were simplified for display to $P < $ 1E-10. When plotting the fold changes of thousands of genes, we used kernel density estimation to create smoothed histograms. To assess the waning of the magnitude of fold changes over time, we used linear mixed effects models with day (centered at day 7) as a continuous fixed effect and gene identifier as a random effect. The slope of the change in absolute value of the log2-fold changes was reported. Data

analysis was performed using R (50) (version 4.4.0) through RStudio version 2024.04.1. Plots were generated using ggplot2 (51) and tidyverse (52).

RNA sequencing data were analyzed as described previously (7). In brief, trimmed reads were aligned to GRCh38 using HISAT2, and the aligned reads were counted using Subread/featureCounts. Differential transcript analysis was performed analogously to that performed for microarrays using limma/voom. Mass spectrometry data were analyzed as described previously (7), using Progenesis LC-Mass Spectrometry software (Nonlinear Dynamics, Newcastle upon Tyne, UK) and the human SwissProt database. We used FDR thresholds of ≤ 0.1 for protein identification and ≤ 0.01 for peptide identification. Protein identification required a minimum of two unique peptides per protein. Following protein identification and quantification, differential protein analysis was performed analogously to that performed for microarrays.

## ACKNOWLEDGMENTS

We wish to express gratitude to all study volunteers for their participation. We acknowledge the Fred Hutch Genomics (Cassie Sather, Crissa Bennett) core facility for their assistance; Ann M. Kahn and Martha Cavallo for their work on MTN-014; Ronit Katz for providing statistical expertise; and Claire Stevens for her assistance.

This work was funded by NIH R01AI116292 (to F.H.), NIH R01AI134293 (to R.M.) and by the National Center for Advancing Translational Sciences of the National Institutes of Health under Award Number KL2 TR002317 (awarded to G.G.G.). M.J.C. and I.M. were supported by the Microbicide Trials Network (UM1AI068633, Sharon Hillier, PI). MTN trials were supported by NIH grants 5UM1A1069466, UM1AI068633, UM1AI068615, and UM1AI106707, with co-funding from the Eunice Kennedy Shriver National Institute of Child Health and Human Development and the National Institute of Mental Health. The content is solely the responsibility of the authors and does not necessarily represent the official views of the National Institutes of Health. The findings and conclusions in this report are those of the authors and do not necessarily represent the official position of the funding agencies. The funders had no role in the study design, data collection and analysis, decision to publish, or preparation of the manuscript. The corresponding author had full access to all the data in the study and had final responsibility for the decision to submit for publication.

## AUTHOR AFFILIATIONS

[1]Department of Obstetrics and Gynecology, University of Washington, Seattle, Washington, USA

[2]Department of Infectious Diseases and Dermatology, University of Barcelona, Barcelona, Spain

[3]Asociación Civil Impacta Salud y Educación, Lima, Peru

[4]Jim Pickett Consulting, Chicago, Illinois, USA

[5]Division of Gastroenterology, Hepatology and Nutrition, University of Pittsburgh School of Medicine, Pittsburgh, Pennsylvania, USA

[6]Magee-Womens Research Institute, Pittsburgh, Pennsylvania, USA

[7]Johns Hopkins University School of Medicine, Baltimore, Maryland, USA

[8]Division of Public Health Sciences, Fred Hutch Cancer Center, Seattle, Washington, USA

[9]Division of Gastroenterology, Department of Medicine, Bronxcare, New York, New York, USA

[10]Department of Epidemiology, ICAP at Columbia University, Columbia University, New York, New York, USA

[11]Department of Medicine, ICAP at Columbia University, Columbia University, New York, New York, USA

[12]Desmond Tutu HIV Centre, Cape Town, South Africa

[13]Centre for Medical Ethics and Law, Department of Medicine, Faculty of Medicine and Health Sciences, Stellenbosch University, Stellenbosch, South Africa

[14]Department of Obstetrics, Gynecology & Reproductive Sciences, Faculty of Medicine, University of Manitoba, Winnipeg, Canada

[15]Department of Pathology, Center for Global Health and Diseases, Case Western Reserve University, Cleveland, Ohio, USA

[16]University of Calgary, Calgary, Alberta, Canada

[17]Department of Obstetrics & Gynecology, Faculty of Medicine, University of Manitoba, Winnipeg, Canada

[18]The Fenway Institute of Fenway Health, Boston, Massachusetts, USA

[19]Fenway Health, Boston, Massachusetts, USA

[20]Division of Infectious Diseases, Department of Medicine, Beth Israel Deaconess Medical Center, Boston, Massachusetts, USA

[21]Department of Epidemiology, University of Washington, Seattle, Washington, USA

[22]Department of Medicine, University of Washington, Seattle, Washington, USA

[23]Department of Global Health, University of Washington, Seattle, Washington, USA

[24]Unit of Infectious Diseases, Department of Medicine Solna, Center for Molecular Medicine, Karolinska Institutet, Stockholm, Sweden

[25]Department of Nutrition, Case Western Reserve University School of Medicine, Cleveland, Ohio, USA

[26]Department of Population and Quantitative Health Sciences, Case Western Reserve University, Cleveland, Ohio, USA

[27]Department of Bioengineering, University of Washington, Seattle, Washington, USA

[28]Vaccine and Infectious Disease Division, Fred Hutch Cancer Center, Seattle, Washington, USA

[29]Department of Medicine, University of Washington, Seattle, Washington, USA

## AUTHOR ORCIDs

Sean M. Hughes http://orcid.org/0000-0002-9409-9405
Claire N. Levy http://orcid.org/0000-0003-3204-211X
Urvashi Pandey http://orcid.org/0000-0002-8993-2642
Ross D. Cranston http://orcid.org/0000-0002-2687-6217
Javier R. Lama http://orcid.org/0000-0002-4983-5725
Craig W. Hendrix http://orcid.org/0000-0002-5696-8665
Mark A. Marzinke http://orcid.org/0000-0003-1670-8786
Adam D. Burgener http://orcid.org/0000-0002-2303-5738
Ian McGowan http://orcid.org/0000-0002-6470-8476
Mark J. Cameron http://orcid.org/0000-0003-4768-4094
Kim A. Woodrow http://orcid.org/0000-0002-9508-8804
Florian Hladik http://orcid.org/0000-0002-0375-2764

## FUNDING

| Funder | Grant(s) | Author(s) |
| --- | --- | --- |
| National Institutes of Health | 5UM1A1069466 | Ian McGowan |
| National Institutes of Health | KL2TR002317 | Germán G. Gornalusse |
| National Institutes of Health | R01AI116292 | Florian Hladik |
| National Institutes of Health | R01AI134293 | Romel Mackelprang |
| National Institutes of Health | UM1AI068615 | Ian McGowan |
| National Institutes of Health | UM1AI068633 | Ian McGowan |
| National Institutes of Health | UM1AI106707 | Ian McGowan |
| National Institutes of Health | R01-AI184122 | Germán G. Gornalusse |

## AUTHOR CONTRIBUTIONS

Sean M. Hughes, Conceptualization, Data curation, Formal analysis, Investigation, Methodology, Project administration, Software, Supervision, Validation, Visualization, Writing – original draft, Writing – review and editing | Fernanda L. Calienes, Investigation | Claire N. Levy, Investigation, Methodology, Writing – review and editing | Urvashi Pandey, Investigation, Methodology | Germán G. Gornalusse, Funding acquisition, Investigation, Methodology, Writing – review and editing | Ross D. Cranston, Conceptualization, Investigation | Javier R. Lama, Conceptualization, Investigation | Jim Pickett, Conceptualization, Investigation | Rhonda M. Brand, Investigation | Craig W. Hendrix, Conceptualization, Investigation | Mark A. Marzinke, Conceptualization | James Y. Dai, Formal analysis | Bhavna Balar, Investigation | Jessica E. Justman, Conceptualization, Funding acquisition, Investigation, Project administration, Writing – review and editing | Gonasagrie Nair, Conceptualization, Investigation, Project administration | Alicia R. Berard, Investigation, Methodology | Kenzie Birse, Investigation, Methodology | Laura Noel-Romas, Investigation | Kenneth H. Mayer, Conceptualization, Investigation, Writing – review and editing | Joanne D. Stekler, Investigation, Writing – review and editing | Romel Mackelprang, Conceptualization, Data curation, Formal analysis, Funding acquisition, Investigation, Methodology, Supervision, Visualization, Writing – review and editing | Adam D. Burgener, Conceptualization, Investigation, Methodology, Supervision, Writing – review and editing | Ian McGowan, Conceptualization, Funding acquisition, Investigation, Project administration, Supervision, Writing – review and editing | Cheryl M. Cameron, Conceptualization, Investigation, Supervision, Writing – review and editing | Mark J. Cameron, Conceptualization, Investigation, Supervision, Writing – review and editing | Kim A. Woodrow, Conceptualization, Writing – review and editing | Florian Hladik, Conceptualization, Data curation, Funding acquisition, Investigation, Methodology, Project administration, Resources, Supervision, Visualization, Writing – review and editing

## DATA AVAILABILITY

The raw microarray data are available at GEO with accession numbers: GSE57025 (MTN-007), GSE277357 (MTN-014), and GSE138723 (MTN-017). The RNAseq data are available at https://doi.org/10.6084/m9.figshare.c.4704827.v2 (MTN-017). The proteomics data are available at https://doi.org/10.6084/m9.figshare.c.7352236.

## ADDITIONAL FILES

The following material is available online.

### Supplemental Material

**File S1 (Spectrum01680-25-s0001.xlsx).** Gene fold changes comparing baseline to treatment within participant.
**File S2 (Spectrum01680-25-s0002.csv).** Results of gene set testing of the Hallmark gene sets.
**File S3 (Spectrum01680-25-s0003.csv).** Results of gene set testing of the Gene Ontology Biological Processes gene sets.
**Supplemental legends (Spectrum01680-25-s0004.docx).** Descriptive legends for Files S1 to S3.
**Table S1 (Spectrum01680-25-s0005.docx).** ISG-24 gene set.

### Open Peer Review

**PEER REVIEW HISTORY (review-history.pdf).** An accounting of the reviewer comments and feedback.

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
