## [Reviewer comments · Microbiology Spectrum]

Microbiology Spectrum

Mucosal tenofovir 1% gel stimulates cell proliferation and type I/III interferon pathways

Sean Hughes, Fernanda Calienes, Claire Levy, Urvashi Pandey, Germán Gornalusse, Ross Cranston, Javier Lama, Jim Pickett, Rhonda Brand, Craig Hendrix, Mark Marzinke, James Dai, Bhavna Balar, Jessica Justman, Gonsagrie Nair, Alicia Berard, Kenzie Birse, Laura Noel-Romas, Kenneth Mayer, Joanne Stekler, Romel Mackelprang, Adam Burgener, Ian McGowan, Cheryl Cameron, Mark Cameron, Kim Woodrow, and Florian Hladik

Corresponding Author(s): Florian Hladik, University of Washington

Review Timeline:

Submission Date:	June 2, 2025
Editorial Decision:	July 29, 2025
Revision Received:	October 8, 2025
Editorial Decision:	January 6, 2026
Revision Received:	January 30, 2026
Accepted:	February 2, 2026

Editor: Vaithilingaraja Arumugaswami

Reviewer(s): Disclosure of reviewer identity is with reference to reviewer comments included in decision letter(s). The following individuals involved in review of your submission have agreed to reveal their identity: Arjit Vijay Jeyachandran (Reviewer #2)

Transaction Report:

DOI: <https://doi.org/10.1128/spectrum.01680-25>

Re: Spectrum01680-25 (Mucosal tenofovir 1% gel stimulates cell proliferation and type I/III interferon pathways)

Dear Dr. Florian Hladik:

Thank you for the privilege of reviewing your work. Below you will find my comments, instructions from the Spectrum editorial office, and the reviewer comments.

Revision Guidelines

Sincerely,
Vaithilingaraja Arumugaswami
Editor
Microbiology Spectrum

Reviewer #1 (Comments for the Author):

The study by Hughes et al. assessed topical tenofovir's effect on the gut and vaginal mucosa over 7, 14 and 56 days in three microbicide trials (MTN-007, MTN-014, and MTN-017) and compared their findings to oral use of TDF/FTC. They found that most gene expression changes reduce over longer periods of gel use, but that stimulation of the type I/III interferon system and of cell proliferation persists. This study was intended just to explain the beneficial or detrimental effect of the drug tenofovir gel in rectum and vagina use in three different time periods. Tenofovir works by inhibiting the activity of viral enzymes, such as reverse

transcriptase in HIV, which are crucial for viral replication. By blocking these enzymes, tenofovir prevents the viruses from making copies of themselves. However, this study highlights only the gene expression changes upon use of Tenofovir gel in different time periods. Since no infection-treated with gel data provided, it is hard to assess the host transcriptional changes and restore the differentially expressed genes upon usage of this gel, which is crucial. This manuscript has 8 figures. It would be better to keep only statistically significant DEGs data in the main manuscript by condensing figures and remove/move others to supplemental. Most of the plots are based on the gene expression data (log₂) without significance, which is not adding value.

Minor comments:

Page 5 - Table 1 does not have the actual number of participants included in each of three clinical studies. It would be easy to read if authors switch the rows based on the treatment length (i.e. 7, 14 and 56 days). Number of samples used from each sample location should be provided.

Figure 1B,C - In these two plots, differentially expressed genes (DEGs) after 7 days of gel use are provided along with the data from 14 and 56 days. It is not clear that all the 1,226 DEGs in 7 days were also differentially expressed (FDR ≤0.05). This needs to be stated clearly.

Table 2 shows some samples collected in vagina even though the treatment location is rectum (vise versa) at 14 days of gel use. It would be interesting to see what happens with 56 days of gel use as authors did not find any DEGs in 14 days. I noticed that 14 days samples were derived only from female subjects. Did authors compare the data derived from both vagina and rectum of the same subjects?

Table 2 - It would be essential to include 7 days data here for easy comparison.

Page 7 / Figure 2 - I am not sure what is the point in plotting the log₂ fold changes with statistically not significant genes. The reliability of the gene expression results should only be based on the statistically significant. The significant gene should only be used for identifying the enrichment of biological processes. The current results do not add values to the field. Can authors show GO Biological process results at least in the supplemental?

Page 8 - Overall the use of gel did alter only a small fraction of transcriptional changes in rectum (56 days; 25/13,249 genes). Besides, RNA-seq is a highly sensitive technique over microarray.

Microarray, RNA-seq and mass spectrometry have their own advantages and limitations. Comparing these findings may not provide correlation.

Figure 4C - I don't see any plot for the left column.

Page 11 - Log₂-fold change >0.25: usually anything less than 1 may be less stringent (chances of false positive). This needs to be clarified.

Table 3 - Move it to supplemental.

Authors referred to their previous manuscript for ethical approval. Did they get one for this study which includes phases II and III?

Page 19 - It would be easy for readers to briefly describe the RNA sequencing and data analysis methods employed in this study. Add FDR next to 'adjusted p-value'.

Reviewer #2 (Comments for the Author):

The manuscript is well-structured and presents important findings that expands our understanding of the mucosal effects of Tenofovir used in HIV prophylaxis. The following comments are offered to further enhance its clarity, precision, and overall presentation:

- In multiple instances (e.g., Figure 1C), the manuscript describes gene expression changes "waning over time." However, statistical testing of this trend (e.g., pairwise comparison across timepoints) is not provided. Consider adding quantitative assessments or clarify that the trends are descriptive.
- Figure 4D, 5B, 6C, 6D: The Hallmark gene set enrichment plots are clear. Consider adding a small legend or footnote to indicate what "Orange" and "Green" signify (upregulated/downregulated) directly on the figures for immediate understanding, even though it is in the main text.
- Table 1: Provides a good overview of study characteristics.
- Consistency in Terminology: Ensure consistent use of "tenofovir 1% gel" versus "tenofovir gel" versus "TFV gel" throughout the manuscript.
- Abbreviations: Ensure all abbreviations are defined upon first use in the abstract and main text (e.g., NRTI, ISG). While mostly consistent, a final check is recommended.

Comments and Suggestions for the Author:

The manuscript is well-structured and presents important findings that expands our understanding of the mucosal effects of Tenofovir used in HIV prophylaxis. The following comments are offered to further enhance its clarity, precision, and overall presentation:

- In multiple instances (e.g., Figure 1C), the manuscript describes gene expression changes “waning over time.” However, statistical testing of this trend (e.g., pairwise comparison across timepoints) is not provided. Consider adding quantitative assessments or clarify that the trends are descriptive.
- Figure 4D, 5B, 6C, 6D: The Hallmark gene set enrichment plots are clear. Consider adding a small legend or footnote to indicate what "Orange" and "Green" signify (upregulated/downregulated) directly on the figures for immediate understanding, even though it is in the main text.
- Table 1: Provides a good overview of study characteristics.
- Consistency in Terminology: Ensure consistent use of "tenofovir 1% gel" versus "tenofovir gel" versus "TFV gel" throughout the manuscript.
- Abbreviations: Ensure all abbreviations are defined upon first use in the abstract and main text (e.g., NRTI, ISG). While mostly consistent, a final check is recommended.

The authors thank both reviewers for their thoughtful reviews. We have responded point-by-point below in green font. The line numbers indicated refer to the manuscript with changes tracked.

Reviewer #1 (Comments for the Author):

The study by Hughes et al. assessed topical tenofovir's effect on the gut and vaginal mucosa over 7, 14, and 56 days in three microbicide trials (MTN-007, MTN-014, and MTN-017) and compared their findings to oral use of TDF/FTC. They found that most gene expression changes reduce over longer periods of gel use, but that stimulation of the type I/III interferon system and of cell proliferation persists. This study was intended just to explain the beneficial or detrimental effect of the use of tenofovir gel in the rectum and vagina in three different time periods. Tenofovir works by inhibiting the activity of viral enzymes, such as reverse transcriptase in HIV, which are crucial for viral replication. By blocking these enzymes, tenofovir prevents the viruses from making copies of themselves.

However, this study highlights only the gene expression changes upon use of Tenofovir gel in different time periods. Since no infection-treated-with-gel data were provided, it is hard to assess the host transcriptional changes and restoration of the differentially expressed genes upon usage of this gel, which is crucial.

- We thank the reviewer for this important point. Our goal was to understand the effect of tenofovir itself on the host in the absence of the virus. Other studies have clearly demonstrated tenofovir's efficacy as prevention and treatment against HIV. We aimed to determine whether this drug, potentially used lifelong, has any side effects on host gene expression. We do acknowledge, though, the reviewer's argument that there may be some variance in how tenofovir affects gene expression in the absence or presence of HIV infection. Therefore, our study findings have their most reliable implications for tenofovir's use as a preventative. We have now added this point to the Limitations section of the Discussion in lines 607-608.

This manuscript has 8 figures. It would be better to keep only statistically significant DEGs data in the main manuscript by condensing figures and remove/move others to supplemental.

- We appreciate this suggestion, but prefer for the figures to remain in the main manuscript, because it makes it much easier for readers to follow the flow of the results. All figures contain panels with statistically significant results.

Most of the plots are based on the gene expression data (log2) without significance, which is not adding value.

- We appreciate this concern. A significant challenge of this paper was the differing sample sizes and, consequently, the statistical power of the included studies. This challenge could result in a gene being statistically different from 0 in one study due to a larger sample size but not significant in another study due to a

smaller sample size, despite having the same or an even larger effect size (fold change). We felt that one way to address this challenge was to consider correlations between effect sizes for selected genes that were statistically different from 0 in at least one study. That approach enabled us to identify similarities between studies of varying sample sizes. It indicated, for instance, that interferon-stimulated genes had similar fold changes across studies, despite being statistically significant in only some studies.

- We certainly understand the limitations of this approach and that, overall, gene expression changes were limited in these studies. Ultimately, this is a reassuring result, given that it indicates limited potential side effects of tenofovir use. We added additional, prominent discussion of these limitations in lines 162-166 and 419-422.

Minor comments:

Page 5 - Table 1 does not have the actual number of participants included in each of the three clinical studies. It would be easier to read if the authors switched the rows based on the treatment length (i.e., 7, 14, and 56 days). The number of samples used from each sample location should be provided.

- Thank you for this suggestion. We have updated Table 1 as suggested.

Figure 1B, C - In these two plots, differentially expressed genes (DEGs) after 7 days of gel use are provided along with the data from 14 and 56 days. It is not clear that all the 1,226 DEGs in 7 days were also differentially expressed (FDR ≤ 0.05). This needs to be stated clearly.

- We have added text clearly indicating how many of those genes were differentially expressed in each study in lines 162-163.

Table 2 shows some samples collected in the vagina, even though the treatment location is the rectum (vice versa), at 14 days of gel use. It would be interesting to see what happens with 56 days of gel use, as the authors did not find any DEGs in 14 days. I noticed that the 14-day samples were derived only from female subjects. Did the authors compare the data derived from both the vagina and the rectum of the same subjects?

- Unfortunately, we only have data from 14 days of vaginal use, as that study ended after 14 days. There are no samples from 56 days of vaginal use.
- The reason that the MTN-014 samples are all from female study participants is that samples were taken from the vagina and the rectum simultaneously from the same participants. The data from the rectum and the vagina come from the same participants.

Table 2 - It would be essential to include the 7-day data here for easy comparison.

- We have updated Table 2 to include the 7 days of treatment arm. Note that in MTN-007, there were 1,254 differentially expressed genes. However, we only compared the expression of 1,226 of those genes to MTN-014 and MTN-017. That is because the remaining 28 genes were not detectable in MTN-014 and/or

MTN-017. This detail is the explanation for the discrepancy between the 1,254 listed in Table 2 and the 1,226 discussed in Figure 1.

Page 7 / Figure 2 - I am not sure what the point is in plotting the log2 fold changes with statistically not significant genes. The reliability of the gene expression results should only be based on the statistically significant.

- We have addressed this point above.

Only the significant genes should be used for identifying the enrichment of biological processes. The current results do not add value to the field.

- We are using a gene set test called “camera” (see publication here: <https://academic.oup.com/nar/article/40/17/e133/2411151>), which is similar in principle to GSEA. This test is designed to take into account the fold changes of every detectable gene (not just differentially expressed genes). The principle is essentially to compare whether the fold changes of a set of genes of interest (e.g., those stimulated in response to interferon) differ in their fold changes from the fold changes of all genes. I.e., do interferon-response genes differ by fold change from what we’d expect at random? A competitive gene set thus utilizes all detectable genes, not just the statistically significant ones.
- This type of test differs from an over-representation test in which only statistically significant genes would be considered.
- In the end, both types of tests would give information about biological processes/gene sets, but they use different data to reach that information.

Can authors show GO Biological process results at least in the supplemental?

- Thank you for this suggestion. We have added a supplemental file with the results of the GO biological processes.

Page 8 - Overall, the use of gel did alter only a small fraction of transcriptional changes in the rectum (56 days; 25/13,249 genes). Besides, RNA-seq is a more sensitive technique than microarray.

- We agree that the gel only altered a small number of genes in MTN-017 at 56 days. For historical reasons, microarrays were used for MTN-007, MTN-014, and MTN-017. In addition, RNAseq was used for MTN-017. We focused on the microarray results and used the RNAseq data as confirmatory results, because microarray was available from all three studies, in contrast to RNAseq.

Microarray, RNA-seq, and mass spectrometry have their own advantages and limitations. Comparing these findings may not provide a correlation.

- We agree that these three techniques have their own advantages and limitations. Indeed, because of the methodological differences between them, we find it highly convincing that the gel-induced changes are real for those genes/proteins for which the disparate technologies agreed (Fig. 3A and B).

Figure 4C - I don't see any plot for the left column.

- Thank you for this comment. We agree that Fig. 4C was confusing. We intended to keep a consistent layout with A and B (significant genes in the plot to the left, non-significant genes in the plot to the right). However, this layout left an awkward blank space with the appearance of something missing by mistake. We have moved the plot in Fig. 4C and added a more explanatory label to that panel.

Page 11 - Log₂-fold change >0.25: usually anything less than 1 may be less stringent (chances of false positive). This needs to be clarified.

- Our goal with this gene set was to include all ISGs potentially affected by oral TDF/FTC, which is why we used such a low fold change cutoff. We have added a sentence to the Results section (lines 389-391) pointing out the possible inclusion of false positives and explaining our rationale.

Table 3 - Move it to supplemental.

- We have changed this table to be Supplemental Table 1.

Authors referred to their previous manuscript for ethical approval. Did they get one for this study which includes phases II and III?

- We have added detailed descriptions of the ethical approval for all studies (including phases II and III) to the Methods section in lines 632-638.

Page 19 - It would be easy for readers to briefly describe the RNA sequencing and data analysis methods employed in this study. Add FDR next to 'adjusted p-value'.

- We have added more detailed descriptions of the wet lab protocols for RNA sequencing and mass spectrometry as well as the data analysis procedures in the Methods in lines 668-676 and 709-717 and have also added "FDR" (lines 693 and 696).

Reviewer #2 (Comments for the Author):

The manuscript is well-structured and presents important findings that expand our understanding of the mucosal effects of Tenofovir used in HIV prophylaxis. The following comments are offered to further enhance its clarity, precision, and overall presentation:

- In multiple instances (e.g., Figure 1C), the manuscript describes gene expression changes "waning over time." However, statistical testing of this trend (e.g., pairwise comparison across timepoints) is not provided. Consider adding quantitative assessments or clarify that the trends are descriptive.
 - Thank you for this suggestion. We have added a quantitative assessment as you suggested. The results of this analysis are described in lines 174-177 and 209-211, and the methods are described in lines 703-706. Briefly, we fit a linear mixed effects model to estimate the rate of change in the gene fold changes vs the number of days of treatment, with the gene identifier as a random effect to account for repeated measurements of the same gene. We confirmed a relatively rapid decay of the genes in Fig. 1C (predicted mean fold change of 0 after just 74 days). In contrast, we found a much slower decay (7.6 times slower) for the genes in Fig. 2B, indicating that the changes to those genes are much more persistent.
- Figure 4D, 5B, 6C, 6D: The Hallmark gene set enrichment plots are clear. Consider adding a small legend or footnote to indicate what "Orange" and "Green" signify (upregulated/downregulated) directly on the figures for immediate understanding, even though it is in the main text.
 - We have added legends to the figures as suggested.
- Table 1: Provides a good overview of study characteristics.
 - We thank the reviewer for this comment
- Consistency in Terminology: Ensure consistent use of "tenofovir 1% gel" versus "tenofovir gel" versus "TFV gel" throughout the manuscript.
 - We have checked the manuscript and made several changes to ensure that we consistently use the phrase "tenofovir 1% gel".
- Abbreviations: Ensure all abbreviations are defined upon first use in the abstract and main text (e.g., NRTI, ISG). While mostly consistent, a final check is recommended.
 - We have checked the manuscript and defined several acronyms for which definitions were missing.

Re: Spectrum01680-25R1 (Mucosal tenofovir 1% gel stimulates cell proliferation and type I/III interferon pathways)

Dear Dr. Florian Hladik:

Thank you for the privilege of reviewing your work. Below you will find my comments, instructions from the Spectrum editorial office, and the reviewer comments.

We had a delay in receiving the second review. The reviewer has raised concerns that all their major critiques were addressed in the study limitation section instead analyzing existing data set. However, they have recommended to address two minor issues before accepting (Editorially).

Please return the manuscript at your earliest or within 60 days; if you cannot complete the modification within this time period, please contact me. If you do not wish to modify the manuscript and prefer to submit it to another journal, notify me immediately so that the manuscript may be formally withdrawn from consideration by Spectrum.

Revision Guidelines

Sincerely,
Vaithi Arumugaswami
Editor
Microbiology Spectrum

Reviewer #1 (Comments for the Author):

The authors addressed most comments through textual revisions and by justifying the lack of additional experiments or analyses. While the limitation in obtaining clinical specimens is understandable, the data generated could still be compared

within the study and against publicly available datasets to strengthen the conclusions.

Not properly addressed previously raised two major comments:

1) "This study highlights only the gene expression changes upon use of Tenofovir gel in different time periods. Since no infection treated with gel data were provided, it is hard to assess the host transcriptional changes and restoration of the differentially expressed genes upon usage of this gel, which is crucial." The authors respond that many previous studies have demonstrated tenofovir's efficacy in HIV prevention and treatment. Given that infected or infected-treated data were not generated in this study, it remains unclear why those existing datasets were not used for comparison with the current results.

2) "Table 2 shows some samples collected in the vagina, even though the treatment location is the rectum (vice versa), at 14 days of gel use. It would be interesting to see what happens with 56 days of gel use, as the authors did not find any DEGs in 14 days. I noticed that the 14 day samples were derived only from female subjects. Did the authors compare the data derived from both the vagina and the rectum of the same subjects?" This point was not addressed. Comparing data from both the vaginal and rectal samples from the same subjects would add value to the study.

Reviewer #2 (Comments for the Author):

This revised manuscript presents technically sound data and employs multiple methodologies that support its conclusions. The study's key strengths lie in its novel findings regarding the persistent induction of type I/III interferon and cell proliferation pathways over extended periods of tenofovir use, and its cross-compartmental analysis revealing unique effects in the vagina. The revised version effectively addresses prior concerns. Overall, this is a valuable contribution to the field.

Reviewer #1 (Comments for the Author):

The authors addressed most comments through textual revisions and by justifying the lack of additional experiments or analyses. While the limitation in obtaining clinical specimens is understandable, the data generated could still be compared within the study and against publicly available datasets to strengthen the conclusions.

Not properly addressed previously raised two major comments:

1) "This study highlights only the gene expression changes upon use of Tenofovir gel in different time periods. Since no infection treated with gel data were provided, it is hard to assess the host transcriptional changes and restoration of the differentially expressed genes upon usage of this gel, which is crucial." The authors respond that many previous studies have demonstrated tenofovir's efficacy in HIV prevention and treatment. Given that infected or infected-treated data were not generated in this study, it remains unclear why those existing datasets were not used for comparison with the current results.

Thank you for your comment.

Oral tenofovir is approved for both HIV prevention and treatment. In contrast, topical tenofovir gel, as studied here, has been tested for prevention but is not approved for that use. In addition, topical tenofovir gel is not intended for treatment of infection; it is only intended for prevention. We may have been unclear about this distinction in our prior response.

To the best of our knowledge, there are no existing gene expression datasets reporting on the effect of topical tenofovir use during HIV infection. Indeed, such datasets will likely never be generated: in the circumstance where someone using topical tenofovir gel contracted HIV, they would immediately be switched to oral treatment and cease topical application.

Moreover, in our study, we specifically sought to understand the effect of topical tenofovir treatment in the absence of infection. Our intention was to determine whether the gel itself induces changes in mucosal gene expression independently of HIV. This information may shed light on the differential efficacy of the topical gel by route of administration (rectal vs vaginal), for instance.

However, we understand the reviewer's desire for even more context. For this reason, a more comprehensive review of the complete prior literature has just been submitted as an invited review article to the American Journal of Reproductive Immunology. This review will provide more background and details. Such a comprehensive review of the literature and synthesis with prior studies is beyond the scope of this primary data paper.

2) "Table 2 shows some samples collected in the vagina, even though the treatment location is the rectum (vice versa), at 14 days of gel use. It would be interesting to see what happens with 56 days of gel use, as the authors did not find any DEGs in 14 days.

I noticed that the 14-day samples were derived only from female subjects. Did the authors compare the data derived from both the vagina and the rectum of the same subjects?" This point was not addressed. Comparing data from both the vaginal and rectal samples from the same subjects would add value to the study.

Thank you for your thoughtful comment.

Regarding 56 days of gel use in the vagina:

Unfortunately, MTN-017, which ran for 56 days, did not include the collection of specimens from the vagina. MTN-014, which ran for 14 days, included tissue specimens from both the vagina and rectum. For that reason, all participants in MTN-014 were female.

Regarding the comparison of vagina and rectum from the same subjects:

In Fig 5b, we compare within-compartment effects: what are the gene set changes in the rectum after rectal treatment and in the vagina after vaginal treatment? As shown in the figure, there is very little overlap in gene set changes between the two sites. We have updated the figure legend for Fig 5b to more clearly state the study arms shown in the figure.

In Fig 6, we compare by route of administration within each tissue:

- Fig 6c: Compare, within the rectum, rectal vs vaginal administration
- Fig 6d: Compare, within the vagina, rectal vs vaginal administration

As shown in the figure, while there is some overlap in gene set changes between rectal and vaginal administration within each site, there is little overlap between the two sites.

Here, we present correlations at the level of individual genes between rectal and vaginal tissues within the same participants:

There are some correlations between the rectum and the vagina when the gel is applied only to the vagina (A; $r=0.14$) or only to the rectum (B; $r=0.11$). Comparing gene changes in the rectum in response to rectal administration with changes in the vagina in response to vaginal administration, there is no correlation (C; $r=-0.004$).

Given the weak correlations and the already considerable length of the paper, we decided not to include these correlation plots. We feel that the gene set data in Figs 5 and 6 are adequate to show the limited similarity between responses in the vagina and rectum.

Reviewer #2 (Comments for the Author):

This revised manuscript presents technically sound data and employs multiple methodologies that support its conclusions. The study's key strengths lie in its novel findings regarding the persistent induction of type I/III interferon and cell proliferation pathways over extended periods of tenofovir use, and its cross-compartmental analysis revealing unique effects in the vagina. The revised version effectively addresses prior concerns. Overall, this is a valuable contribution to the field.

Thank you for your comments.

Re: Spectrum01680-25R2 (Mucosal tenofovir 1% gel stimulates cell proliferation and type I/III interferon pathways)

Dear Dr. Florian Hladik:

Congratulations! Your manuscript has been accepted, and I am forwarding it to the ASM production staff for publication. Your paper will first be checked to make sure all elements meet the technical requirements. ASM staff will contact you if anything needs to be revised before copyediting and production can begin. Otherwise, you will be notified when your proofs are ready to be viewed.

Sincerely,
Vaithi Arumugaswami
Editor
Microbiology Spectrum